# Integrating Rankings into Quantized Scores in Peer Review

**Yusha Liu**                                                    *yushal@cs.cmu.edu*
*Machine Learning Department*
*Carnegie Mellon University*

**Yichong Xu**                                             *yichong.xu@microsoft.com*
*Microsoft Cognitive Services Research*

**Nihar B. Shah**                                                *nihars@cs.cmu.edu*
*Machine Learning Department, Computer Science Department*
*Carnegie Mellon University*

**Aarti Singh**                                                   *aarti@cs.cmu.edu*
*Machine Learning Department*
*Carnegie Mellon University*

**Reviewed on OpenReview:** *https://openreview.net/forum?id=Kb1lbOvSLa*

## Abstract

In peer review, reviewers are usually asked to provide scores for the papers. The scores are then used by Area Chairs or Program Chairs in various ways in the decision-making process. The scores are usually elicited in a quantized form to accommodate the limited cognitive ability of humans to describe their opinions in numerical values. It has been found that the quantized scores suffer from a large number of ties, thereby leading to a significant loss of information. To mitigate this issue, conferences have started to ask reviewers to additionally provide a ranking of the papers they have reviewed. There are however two key challenges. First, there is no standard procedure for using this ranking information and Area Chairs may use it in different ways (including simply ignoring them), thereby leading to arbitrariness in the peer-review process. Second, there are no suitable interfaces for judicious use of this data nor methods to incorporate it in existing workflows, thereby leading to inefficiencies.

We take a principled approach to integrate the ranking information into the scores. The output of our method is an updated score pertaining to each review that also incorporates the rankings. Our approach addresses the two aforementioned challenges by: (i) ensuring that rankings are incorporated into the updated scores in the same manner for all papers, thereby mitigating arbitrariness, and (ii) allowing to seamlessly use existing interfaces and workflows designed for scores. We empirically evaluate our method on synthetic datasets as well as on peer reviews from the ICLR 2017 conference, and find that it reduces the error by approximately 30% as compared to the best performing baseline on the ICLR 2017 data.

## 1   Introduction

Many applications involve people evaluating a large number of items in a distributed fashion. An important and prominent such application, which is the focus of this paper, is peer review of papers in scientific conferences. A typical way of collecting reviews is through quantized scores, that is, where reviewers are asked to provide scores from a constant number of quantization levels. For example, reviewers for conferences are often asked to provide their opinions on papers in the format of five-level or ten-level Likert items, which are used to evaluate the qualities of submitted papers.

A drawback of such quantized scores is that there exist a large number of ties in such quantized scores, due to the constant number of quantization levels as well as the respondents' tendency to give equal values in

the absence of prompt for relative relationships (Feather, 1973). For example, a recent study (Shah et al., 2018) of peer-review data from the NeurIPS 2016 conference found that among all instances where a reviewer reviewed a pair of papers, the pair of review scores provided by the reviewer were tied in more than 30% of such instances. Such a large proportion of ties were found to exist in the scores for all four criteria elicited from reviewers, and the number of ties was even higher when restricting attention to only the top and middle-quality papers. Such ties result in a loss of information, thereby contributing to the difficulty in making decisions regarding the acceptance of papers.

Despite the apparent drawbacks, there are strong reasons that quantized scores are a widely used format instead of continuous-valued scores to elicit the opinions of reviewers. This is because of the limited ability of humans to describe stimuli with numerical values (Miller, 1956). Psychometric studies have discussed the appropriate number of response alternatives when eliciting responses from humans and suggested that it remain a small constant (Lietz, 2010; Jones & Loe, 2013).

An alternative form of evaluation comprises rankings. Ranking information alone has been demonstrated to be a robust way to collect information (Rankin & Grube, 1980; Douceur, 2009; Shah et al., 2013; 2016b). For example, crowdsourcing experiments (Shah et al., 2016b) demonstrate the reliability of answers in form of pairwise comparisons, which incur a lower per-sample error as compared to eliciting numerical values.

Scientific conferences, which face a growing number of submissions each year, are increasingly trying to collect additional information by asking reviewers to report rankings in addition to scores (Soergel et al., 2013). Among recent computer science conferences, the NeurIPS 2016 conference asked each reviewer to also rank the papers they were reviewing (Shah et al., 2018). This was an experiment that provided a sanity check about rankings in peer review and also identified several benefits of collecting rankings. The ICML 2021 conference, as well as the recent AAAI 2023 conference collected rankings in addition to scores from reviewers on their assigned papers. Note that during the review process, a majority of the reviewers' time is spent on reading and evaluating the papers. As a result, generating rankings in addition to the traditional scores may take only little additional time.

In Figure 1a we illustrate the standard interface used by Area Chairs in peer review, augmented with the ranking information (in the "rankings" column). An important aspect of the standard interface, extensively used by chairs in their workflow, is the ability to sort papers via the minimum, maximum, or average received scores or via the spread of the scores. The augmented ranking information shows the (partial or total) rankings provided by reviewers, where papers not being handled by that Area Chair are replaced with a $*$ symbol (as done by ICML 2021). There are multiple challenges of such an interface. First, such ranking information is incompatible with commonly used workflow elements such as sorting according to scores, so any such operations will omit ranking information. Second, it is not clear how to efficiently extract information from the rankings under such an cluttered interface. As a consequence, the varying use or lack of use of ranking information by different Area Chairs can add to the arbitrariness in peer review. Thus, while the elicitation of ranking evaluations introduces another kind of information for chairs to use, the question of how to judiciously use this data has remained open.

Our goal is to allow the chairs to use the ranking information in addition to the scores while not disrupting their workflow. To this end, we design a method to integrate the reviewer-provided rankings into the scores. One may think of this approach as dequantizing the scores. In our problem formulation (Section 3), the output is a real-valued score for each review, where these real-valued scores combine the reviewer-provided (quantized) scores and rankings. We refer to these real-valued outputs as the "dequantized scores" and illustrate the interface with these dequantized scores in Figure 1b. As shown in Figure 1b, these dequantized scores, unlike the rankings, can now be incorporated in the standard workflow for Area Chairs or Program Chairs, allowing them to seamlessly perform the tasks that they conventionally perform on the original quantized scores (such as sorting by the average or the spread of the scores for each paper).

**Our contributions.** The main contributions of this paper are as follows.

(1) We identify and formulate the problem of combining rankings and quantized scores, focusing specifically on dequantization of the scores for every review, rather than aggregating the quantized scores and/or rankings given by reviewers to estimate a score for each paper. We provide detailed motivations for dequantizing the

| Paper ID | Title | Reviewers | Scores | Rankings | Overall Evaluation (Scores) | | | |
|---|---|---|---|---|---|---|---|---|
| | | | | | Min | Max | Avg | Spread |
| e.g. <3 *Clear* | filter... *Clear* | filter... *Clear* | filter... *Clear* | filter... *Clear* | e.g. <3 *Clear* | e.g. <3 *Clear* | e.g. <3 *Clear* | e.g. <3 *Clear* |
| 134 | The Hobbit | R1; R3; R5 | 6; 7; 7 | [R1]: * > 134
[R4]: * > 134, * > 6390 | 6 | 7 | 6.67 | 0.47 |
| 699 | The Fellowship of the Ring | R1; R5; R8 | 5; 5; 5 | [R8]: * > * > 699 | 5 | 5 | 5 | 0 |
| 1209 | The Two Towers | R2; R5; R7 | 6; 6; 6 | [R2]: 1209 > * | 6 | 6 | 6 | 0 |
| 6390 | The Return of the King | R4; R5; R6 | 7; 6; 7 | [R4]: * > 134, * > 6390
[R5]: 6390 > *
[R6]: * > * > 6390 > * | 6 | 7 | 6.67 | 0.47 |

(a) Interface with quantized scores and rankings.

| Paper ID | Title | Reviewers | Original Scores | Dequantized Scores | Overall Evaluation (Dequantized Scores) | | | |
|---|---|---|---|---|---|---|---|---|
| | | | | | Min | Max | Avg | Spread |
| e.g. <3 *Clear* | filter... *Clear* | filter... *Clear* | filter... *Clear* | filter... *Clear* | e.g. <3 *Clear* | e.g. <3 *Clear* | e.g. <3 *Clear* | e.g. <3 *Clear* |
| 134 | The Hobbit | R1; R3; R5 | 6; 7; 7 | 5.77; 6.54; 6.85 | 5.77 | 6.85 | 6.39 | 0.46 |
| 699 | The Fellowship of the Ring | R1; R5; R8 | 5; 5; 5 | 4.86; 5.17; 5.10 | 4.68 | 5.17 | 4.98 | 0.22 |
| 1209 | The Two Towers | R2; R5; R7 | 6; 6; 6 | 6.27; 6.18; 6.48 | 6.18 | 6.48 | 6.31 | 0.13 |
| 6390 | The Return of the King | R4; R5; R6 | 7; 6; 7 | 7.10; 6.02; 7.01 | 6.02 | 7.10 | 6.71 | 0.49 |

(b) Interface with dequantized scores.

Figure 1: An illustration of the envisaged interfaces for the Area Chairs, a conference management system (Microsoft CMT) used commonly by computer science conferences for peer-review. The top figure shows the interface with separate quantized scores and rankings reported by reviewers. The rankings in the top figure are processed in the same manner as in the ICML 2021 conference, precisely, papers that are not handled by an Area Chair are replaced with ⋆ symbols for confidentiality. The bottom figure shows the interface with our proposed output format of dequantized scores. Each row represents a paper that the Area Chair is handling, while the columns correspond to relevant information of the paper.

scores in Section 4.1. Our work is applicable even if each reviewer reviews only a small subset of the papers, and also if reviewers may only return a partial ranking of the assigned papers. For concreteness, we focus on the application of peer-review, while noting that the problem and solution may also apply to other settings where both rankings and quantized scores are available.

(2) We propose a computationally-efficient algorithm that outputs a real-valued dequantized score for every review in the assignment.

- Our approach is based on a set of design principles we outline subsequently. We make no parametric assumptions on the data generation process, nor the existence of ground truth quality scores and global rankings.

- A part of our approach is inspired by isotonic regression, and we also provide connections to estimation under the Thurstone model and the balanced-rank-estimation algorithm (Wauthier et al., 2013, Section 4.1).

- Our algorithm includes a purely data-driven "Quantization Validation (QV)" method for hyperparameter selection due to the absence of the ground truth.

(3) We evaluate the empirical performance of our proposed algorithm on both synthetic data and a semi-synthetic dataset based on real review scores from the ICLR 2017 conference. As motivated subsequently in Section 4.2, we use the (Kendall-tau) ranking error as the metric of interest. We compare the proposed

algorithm to two baselines: quantized scores and BRE-adjusted-scores, which are formally introduced in Section 5.2. We find that our algorithm incurs about 28% lower error as compared to the best performing baseline in the ICLR 2017 data.

## 2 Related Work

There is a long line of literature (Wauthier et al., 2013; Eriksson, 2013; Braverman et al., 2016; Shah et al., 2016a; Shah & Wainwright, 2017; Shah et al., 2016b; Mao et al., 2017; Negahban et al., 2017; Pananjady et al., 2017; Agarwal et al., 2018; Makhijani & Ugander, 2019; Wang et al., 2020) in the domain of ranking from pairwise comparisons with various modeling assumptions. Methods from this domain take comparison outcomes between pairs of items as input, and output estimates of an underlying global ranking or a comparison probabilities matrix. The problems investigated in this domain, however, are fundamentally different from ours with major distinctions in both the input and output of the problem. In our setting, the algorithm needs to effectively incorporate scores in addition to rankings, while the above methods cannot be trivially adapted to take score values as input. In settings such as peer review, scores cannot be neglected, as the ranking information is very limited and sparse. Additionally, the number of comparisons per item needed in most prior work in ranking from pairwise comparisons grows with the number of items (sometimes logarithmically but often linearly). In practice, however, there are often only 3 to 6 reviews per paper in most peer-reviewed conferences, and the number of comparisons per paper is a small constant. Mao et al. (2017) allow for the number of comparisons to be constant, but still use at least tens of comparisons per item in their simulations, which remains impractical for the peer-review setting. Furthermore, unlike their goals, we make a deliberate design choice to *not* aim to output global ranking or comparisons probabilities of the papers, but instead focus on estimation of dequantized scores for reviewer-paper pairs. This design choice is motivated in Section 4.1.

Some recent works (Hopkins et al., 2019; Zeng & Shen, 2020; Hopkins et al., 2020; Xu et al., 2020a; 2017; 2020b;c; Somers et al., 2017) develop approaches to use ranking information in addition to labels (scores) in supervised tasks such as classification, regression, and optimization. However, there are crucial differences between their settings and ours, such as the preservation of distinct reviewer evaluations, instead of pooling the data together, which prohibit any direct translations of those works to our setting. These works consider the generalization setting by building general predictive models from training data that then apply to incoming test data. Whereas our work is in the transduction setting, where we derive the output scores directly from the input quantized scores and rankings.

The kind of data considered in the paper by Ailon (2010) is closer to our setting despite their goal of deriving a global ranking, which considers two kinds of input: a total ranking for only a few top-ranked items and quantized scores. However, the method is comparison-based and therefore treats the quantized scores as (partial) rankings. This step thus leads to a significant loss of information. For instance, given a total of two papers $X$ and $Y$, a reviewer giving a score 8 to paper $X$ and score 3 to paper $Y$ is very different from giving 4 to $X$ and 3 to $Y$, but the model would only retain the relative ranking between $X$ and $Y$ and not distinguish these two cases. In our setting, such data is equivalent to having only partial rankings but no scores.

The paper most closely related to our setting is a concurrent and independent work by Pearce & Erosheva (2022), which also considers a transduction setting with both ranking and score data. They propose a model termed the Mallows-Binomial model, parametric model that jointly captures the scores and rankings. They estimate the model parameters via maximum likelihood estimation and propose two computationally-efficient algorithms based on A* tree search. They present theoretical results including properties of the Maximum likelihood estimator (MLE) for model parameters. On the empirical front, they fit the model on a real-world grant panel review dataset where 6 judges each scored all of the 18 proposals and ranked their top 6. They examine the results manually in absence of ground truth and show that the estimated model parameters successfully capture information from both scores and rankings.

The work of Pearce & Erosheva (2022), however, differs from ours in several critical ways. First, their method and analysis require that each reviewer reviews the entire set of objects to be evaluated. While this condition may be met in small grant proposal panels such as the American Institute of Biological Sciences

grant proposal review studied in Pearce & Erosheva (2022), this is impractical in conference peer-review that has thousands of submitted papers, which is the primary focus of our work. Second, they assume that each reviewer provides a top-K ranking. We allow reviewers to provide comparisons between any arbitrary subsets of assigned papers (Section 3), which eventually constitutes a partial ranking different from that in Pearce & Erosheva (2022). Third, the Mallows-Binomial model assumes the existence of a true underlying quality score for each paper and consequently of a global ranking of the papers, whereas our approach deliberately aims to avoid assumptions about the ground truth scores and about ranking over the papers. Fourth, their model assumes that rankings and quantized scores are independent conditioned on the model parameters. On the contrary, we assume that the rankings provided by any reviewer are consistent – and hence strongly dependent – with the quantized scores provided by that reviewer. Our setting occurs more naturally in peer-review, where reviewers report the two types of information at the same time as their final evaluations. Fifth, they pool the reviewers' data together to obtain a final quality score for each paper. On the other hand, our approach seeks to provide flexibility to the Area Chairs and Program Chairs on aggregating the reviews for any paper, and hence we output a score for each review.

Finally, there has been a flurry of works that address various other problems in peer review, such as bias (Tomkins et al., 2017; Stelmakh et al., 2019a; Manzoor & Shah, 2021), miscalibration (Flach et al., 2010; Roos et al., 2011; Wang & Shah, 2019; Tan et al., 2021), subjectivity (Lee, 2015; Noothigattu et al., 2021), dishonest behavior (Vijaykumar, 2020; Littman, 2021; Wu et al., 2021; Jecmen et al., 2020; Xu et al., 2019; Dhull et al., 2022), and others. See Shah (2022) for a survey. In particular, Noothigattu et al. (2021) address the problem of reviewer subjectivity, where their proposed output is similar to that we consider here – an updated score pertaining to each review.

## 3 Problem Setting

Consider a set of $P$ papers indexed by $p \in [P]$, and a set of $R$ reviewers indexed by $r \in [R]$.[1] Let $\mathcal{A}$ denote the set of assigned reviewer-paper pairs: $\mathcal{A} = \{(r, p) \in [R] \times [P] \mid \text{reviewer } r \text{ reviews paper } p\}$. For every paper $p$ assigned to reviewer $r$, the reviewer reports a quantized score $z_{rp} \in \mathbb{Z}$ bounded by a pre-specified interval. In this work, we assume that a higher score value indicates a more favorable review. In our experiments, both the number of papers assigned to a reviewer and the number of reviews received by a paper are constants, which is the case in practice in peer-review, as opposed to scaling with $P$ or $R$.

In addition, the reviewer reports a partial ranking $\pi_r$, which is a partial ordering over the set of papers that is assigned to this reviewer. For a pair of papers $(p, p')$ reviewed by reviewer $r$, we use $p \succ_r p'$ to mean that reviewer $r$ ranks paper $p$ above paper $p'$ in the partial ranking $\pi_r$ provided by reviewer $r$. The partial ranking $\pi_r$ can be then represented as the set $\pi_r = \{(p, p') : (r, p) \in \mathcal{A}, (r, p') \in \mathcal{A}, p \succ_r p'\}$. We assume that for every reviewer, the reported rankings are consistent with the scores, that is, the reviewer may give the ranking $p \succ_r p'$ only when the reported scores satisfy $z_{rp} \geq z_{rp'}$.

Observe that under our problem formulation, reviewers can choose to report a total ranking of assigned papers, but also have the freedom to not compare some pairs of papers. In peer review, it is not always reasonable to ask for a binary comparison between every pair of papers. For example, as pointed out in Rogers & Augenstein (2020), it can be difficult to make a comparison in a situation with "incomplete evaluation in one borderline paper vs narrow applicability of another". Allowing partial orderings prevents reviewers from being forced to make such "apples-to-oranges" comparisons (Rogers & Augenstein, 2020).

We now describe our objective given the inputs of quantized scores and rankings. As introduced in Section 1 and detailed in Section 4.1, given the quantized scores and rankings from the reviewers, our goal is to merge the two sources of information into a single source, in a manner that can seamlessly assist the human experts. Our designed algorithm thus aggregates the reviewer-provided sources of information into dequantized and continuous scores. For every $(r, p) \in \mathcal{A}$, we denote the output dequantized score as $\widehat{y}_{rp} \in \mathbb{R}$.

---

[1]For any positive integer m, we use the standard notation $[m] = \{1, 2, \ldots m\}$.

## 4  Main Results

We first describe our three design principles in Section 4.1. Then in Section 4.2, we present our proposed algorithm designed based on these principles. In Section 4.3, we draw interesting connections between our algorithm and simpler algorithms under some extreme cases, as well as a connection to maximum likelihood estimation under the Thurstone model. Finally, we present our data-driven 'Quantization Validation' method for hyperparameter selection in Section 4.4.

### 4.1  Approach and Design Principles

In this section, we introduce three principles we follow in designing our algorithm.

The first principle pertains to our approach towards the goal of the algorithm. Unlike many prior works analyzing scores and ranking information which focus on estimation of the "true" underlying qualities of each item (Volkovs & Zemel, 2012; Shah et al., 2016b; Hajek et al., 2014; Negahban et al., 2017; Wang et al., 2020) or the ranking of such assumed "true" qualities (Braverman & Mossel, 2007; Shah et al., 2016a; Shah & Wainwright, 2017; Pananjady et al., 2017; Volkovs & Zemel, 2012; Mao et al., 2017), we deliberately focus on obtaining a dequantized numerical output corresponding to *each review*. This is because, in the actual peer-review process, the final aggregation depends on various additional factors including the review and rebuttal text, reviewer discussions, and Program/Area Chairs' preferences on how to aggregate individual reviews. The reviewer-provided scores are usually used heavily in the peer-review workflow for sorting or initial judgments but do not fully determine the final decision. Furthermore, we deliberately do *not* want to base our algorithm on the assumption of the existence of some objective true qualities, especially in applications such as peer review that has a significant amount of subjective opinion (Cortes & Lawrence, 2021). Thus with our goal of helping the Program Chairs and Area Chairs seamlessly make their decisions in the real world, we focus on producing a unified representation in form of dequantized scores $\{\widehat{y}_{rp}\}_{(r,p)\in\mathcal{A}}$. This approach of computing updated scores for each review is also taken by Noothigattu et al. (2021) in their work on mitigating subjectivity in peer review.

**Design Principle 1** (Dequantization of scores and not final decisions)**.** *The goal is to produce estimated dequantized scores $\{\widehat{y}_{rp}\}_{(r,p)\in\mathcal{A}}$, and* not *final decisions on the set of papers.*

There are two ways to interpret the dequantized scores $\{\widehat{y}_{rp}\}_{(r,p)\in\mathcal{A}}$, one under a model-based setting and another that does not rely on modeling assumptions.

(1) Under a model-based setting, outcomes in form of partial rankings and scores are generated from latent and real-valued variables. For example in the commonly-used Thurstone model (Thurstone, 1927), the latent variable $y_{rp}, (r,p) \in \mathcal{A}$ represents the inherent opinion of reviewer $r$ for paper $p$ and is drawn from a normal distribution. Under this modeling assumption, $\{\widehat{y}_{rp}\}_{(r,p)\in\mathcal{A}}$ can be interpreted as estimations of ground-truth values $\{y_{rp}\}_{(r,p)\in\mathcal{A}}$. In other words, it is the scores that reviewers would have given *without* the quantization process, which reflect the "true" opinion of the reviewers. This model aligns with certain models proposed in the literature: the model considered in Section 3.2 of Zarkoob et al. (2022) and model number 2 in Heyard et al. (2022) also assume a real-valued latent score per reviewer-paper pair, that yields the quantized score.

(2) The assumption that reviewers generate some unquantized scores in their mind before giving a (quantized) score, as assumed by models such as the Thurstone model, may not always hold in practice. As discussed earlier in Section 1, people may not be capable of evaluating items with fined-grained measure scales (Miller, 1956; Jones & Loe, 2013; Lietz, 2010; Shah et al., 2016b). In this case, one can alternatively consider the dequantized scores $\{\widehat{y}_{rp}\}_{(r,p)\in\mathcal{A}}$ as a set of continuous values that integrate the information from quantized scores and rankings *without* assuming the existence of ground-truth scores. The values of these variables are still compatible with the traditionally score-based decision rules, and other workflow elements used by Area and Program Chairs (Figure 1).

With this motivation, we now present our second principle which requires the estimates to be consistent with the input. In this work, we assume that any integer $z$ represents real values in the interval from $z - 0.5$ and $z + 0.5$, which we refer to as quantization interval of $z$.

**Design Principle 2** (Consistency)**.** *The values of dequantization output $\{\widehat{y}_{rp}\}_{(r,p)\in\mathcal{A}}$ (i) must be consistent with reviewer-provided quantized scores, that is, the dequantized value must be inside the quantization interval associated with the quantized score, $z_{rp} - 0.5 \leq \widehat{y}_{rp} \leq z_{rp} + 0.5$, for every $(r,p) \in \mathcal{A}$; and (ii) must also be consistent with reviewer-provided rankings, that is, the ordering of $\{\widehat{y}_{rp}\}_{p:(r,p)\in\mathcal{A}}$ strictly respects $\pi_r$, for every reviewer $r$.*

The above design principles leave us with many possible choices for $\widehat{y}_{rp}, (r,p) \in \mathcal{A}$. For instance, consider a scores-only setting where we observe the quantized score $z_{rp}$. For a reviewer-paper pair $(r,p) \in \mathcal{A}$, the quantized score $z_{r,p}$ defines only the range of possible values for the dequantized score $\widehat{y}_{r,p}$. Based on only the principles we have established so far, all output values $\widehat{y}_{rp} \in [z_{rp} - 0.5, z_{rp} + 0.5]$ are equally good. Consequently, we establish another design principle to break ties within this interval.

We use the consensus value among reviewers as the additional signal to break ties for quantized scores. The consensus value for a paper is defined as the average of the scores provided by the reviewers for that paper. Based on only the information from the quantized scores given by other reviewers for that paper, it is natural to envisage that the dequantized score lies closer to the consensus value than away from it. In other words, the consensus signal indicates part of the interval $[z_{r,p} - 0.5, z_{r,p} + 0.5]$ in which the dequantized score should lie. For example, without any ranking information, if a reviewer $r$ gave a score of 7 and all the other reviewers gave 7 as well for paper $p$, then the consensus signal from other reviewers offers no additional information within the interval $[6.5, 7.5]$. However, if the other reviewers all gave quantized scores 4, then their consensus signal indicates that the dequantized score $y_{r,p}$ should lie within $[4, 7] \cap [7 - 0.5, 7 + 0.5]$, which is $[6.5, 7]$.

**Design Principle 3** (Consensus)**.** *We break ties in scores due to quantization in the direction of reviewer consensus.*

How much does the dequantized score move in the direction of consensus? This is governed by a hyperparameter in our algorithm, whose value is chosen in a data-dependent manner.

## 4.2 Proposed algorithm

Our three design principles then lead to our proposed algorithm, Algorithm 1. The algorithm takes both reviewer-provided rankings and quantized scores as inputs, and outputs dequantized scores $\{\widehat{y}_{rp}\}_{(r,p)\in\mathcal{A}}$ (Design Principle 1). The algorithm solves a constrained convex optimization problem. The two constraints ensure that the output values are consistent with the reviewer-provided information (Design Principle 2). The term $\left( y_{rp} - \frac{1}{|\{r':(r',p)\in\mathcal{A}\}|} \sum_{r':(r',p)\in\mathcal{A}} y_{r'p} \right)^2$ in the objective measures disagreement between reviewers, therefore capturing the consensus (Design Principle 3). This term and the term $(y_{rp} - z_{rp})^2$ together capture the move towards the consensus, and the amount of movement is determined by a hyperparameter $\lambda > 0$. In Section 4.4, we introduce a proposed cross-validation-like procedure called Quantization Validation for selecting $\lambda$.

---

**Algorithm 1** Proposed algorithm

---

*Inputs:* Quantized scores $\{z_{rp}\}_{(r,p)\in\mathcal{A}}$ and rankings $\{\pi_r\}_{r\in[R]}$ given by reviewers, hyperparameter $\lambda$, small value $\epsilon$

*Pre-processing:* Remove inconsistency $\pi_r = \pi_r \setminus \{(p,p') : (r,p) \in \mathcal{A}, (r,p') \in \mathcal{A}, p \succ_r p' \text{ and } z_{rp} < z_{rp'}\}, \forall r \in [R]$

*Output:* Solution of the following constrained convex optimization program:

$$\underset{\{y_{rp}\}_{(r,p)\in\mathcal{A}}}{\arg\min} \left( \sum_{p\in[P]} \sum_{r:(r,p)\in\mathcal{A}} \left( y_{rp} - \frac{1}{|\{r':(r',p)\in\mathcal{A}\}|} \sum_{r':(r',p)\in\mathcal{A}} y_{r'p} \right)^2 + \lambda \sum_{(r,p)\in\mathcal{A}} (y_{rp} - z_{rp})^2 \right),$$

such that $y_{rp} \geq y_{rp'} + \epsilon$ whenever $(r,p) \in \mathcal{A}, (r,p') \in \mathcal{A},$ and $p \succ_r p'$;

$$z_{rp} - 0.5 \leq y_{rp} \leq z_{rp} + 0.5 \quad \forall (r,p) \in \mathcal{A}.$$

---

It remains to specify the parameter $\epsilon$, which can simply be chosen to be a small positive constant. This parameter is only meant to ensure that any partial ranking given by a reviewer is strictly followed. We pick $\epsilon = 0.05$ in our subsequent experiments while noting that our results are robust to the choice of small $\epsilon$ (Appendix A.4). Our algorithm is inspired in part by isotonic regression (Barlow et al., 1972): if $\epsilon = 0$, then the term $(y_{rp} - z_{rp})^2$ in the objective and the resulting constraints $y_{rp} \geq y_{rp'} \quad \forall \, (r, p) \in \mathcal{A}, (r, p') \in \mathcal{A}$, simply represent isotonic regression to incorporate the rankings.[2]

**Pre-processing** In practice, a reviewer might make mistakes when interacting with the interface and give rankings inconsistent with the scores. Practitioners can use a simple pre-processing step to resolve the inconsistency. The pre-processing step removes from the set of reviewer-provided rankings (Section 3) the pairwise rankings that are inconsistent with quantized scores: $\pi_r = \pi_r \setminus \{(p, p') : (r, p) \in \mathcal{A}, (r, p') \in \mathcal{A}, p \succ_r p' \text{ and } z_{rp} < z_{rp'}\}$.

**The optimization problem.** When $\lambda > 0$, the objective is *strictly convex*. Note that the convexity of the first term follows from the fact that this term is a sum over functions, where each function is a composite of a convex function and an affine function. The set of constraints are linear inequalities, therefore the optimization problem yields a *unique solution*. Specifically, it is a convex quadratic programming (QP) problem, and therefore can be solved in time polynomial in the number of reviewers and papers.

**Using the outputs: Scores or percentiles.** We can extract two kinds of information from the output of the algorithm to present to the Program Chairs and/or Area Chairs. The dequantized scores for all reviewer-paper pairs in the assignment $\{\widehat{y}_{rp}, (r, p) \in \mathcal{A}\}$ themselves provide a convenient and easy-to-use interface that is much cleaner than showing the quantized scores together with raw rankings, as demonstrated in Figure 1. The scores produced by the proposed algorithm lie within the quantization interval associated with the quantized scores (Design Principle 2). Within the intervals, the scores are in arbitrary scale because determining their exact values would essentially require more stringent modeling assumptions than we make in this work. The use of arbitrary scales is not new and is employed in various other applications to express measurements in absence of absolute values (Adinolfi & Sargent, 2017; Bucher et al., 2006; Kamat, 2019; Hoffman, 2004).

An alternative useful type of information that can be derived from this data is the ranking or percentile of each score $\widehat{y}_{rp}$, which represents the relative position of evaluation given by $r$ to $p$ among the entire pool of all review scores (across all assigned reviewer-paper pairs). In Figure 1b, the percentiles can be shown in place of the dequantized scores and enjoy the same compatibility as dequantized scores with the workflow of the chairs. Either of these types of information can then be provided to an Area Chair or Program Chair to help inform decisions across a wider scope of papers that are not accessible to individual reviewers.

In the sequel, we primarily focus on evaluating the performance in terms of ranking error of the scores for all reviewer-paper pairs. The Kendall-tau ranking error we use is formally defined in Section 5.1.

### 4.3 Analysis of the proposed algorithm in special cases

In this section, we analyze several properties of Algorithm 1. First, we draw a connection to Balanced Rank Estimation (BRE) algorithm from Wauthier et al. (2013) (Section 4.3.1) under certain assumptions. We then characterize the output by giving its analytical solution when only scores are present (Section 4.3.2). We also make a connection to the Thurstone model with quantization under the score-only case (Section 4.3.3).

### 4.3.1 Connection to Balanced Rank Estimation (BRE) when $\lambda = \infty$

Balanced Rank Estimation (BRE) (Wauthier et al., 2013, Section 4.1) is an algorithm for rank recovery with pairwise comparisons which enjoys optimal theoretical guarantees under the standard comparison-only setting, under three assumptions: there is a ground-truth global ranking, the pairs compared are chosen independently and uniformly at random, and each pairwise comparison has an identical noise distribution.

---

[2]Note that isotonic regression itself cannot break ties between quantized scores for each reviewer.

In a nutshell, BRE first estimates a score for each item. The estimated score is proportional to the difference between the number of items preceding and succeeding this item in the given pairwise comparisons. Then, BRE uses these estimated scores to induce a global ranking of all items and the global ranking is the final output of their method.

In what follows, we show the connection between our algorithm and BRE under a certain setting. In this setting: (i) every reviewer gives a total ranking of all the papers assigned to them, and (ii) $\lambda = \infty$ in our algorithm, that is, we retain only the second term in the objective in Algorithm 1. Without the first term in its objective, the optimization problem in Algorithm 1 can be solved separately and in closed-form for each reviewer, and simply corresponds to the problem of breaking ties amongst papers with the same score using comparisons. For a reviewer-paper pair $(r, p) \in \mathcal{A}$, the output $\widehat{y}_{rp}$ equals the sum of quantized score $z_{rp}$ given by the reviewer and an arbitrary-scale value generated from the ranking $\pi_r$. We next describe the connection between this ranking-induced value and the output of the BRE algorithm. Consider the set of pairwise comparisons indicated by the total ranking $\pi_p$, and further restrict our attention to the subset of comparisons between paper $p$ and papers with the same quantized scores as $p$. The ranking-induced value is then proportional to the difference between the number of papers preceding and succeeding $p$ in this subset of comparisons. This is the same way that the BRE algorithm estimates the score for paper $p$ given the subset of comparisons as input.

The following proposition characterizes the behavior of the algorithm in the limit of $\lambda = \infty$.

**Proposition 1.** *Assume that reviewers give total rankings of assigned papers, and that $\epsilon$ is a small constant such that the program in Algorithm 1 is feasible. When $\lambda = \infty$, our algorithm generates scores separately across different reviewers. For each reviewer, our algorithm adds to the quantized scores values that are proportional to the estimated scores from the Balanced Rank Estimation (BRE) (Wauthier et al., 2013) algorithm.*

Therefore, we refer to the special case which our algorithm reduces to when $\lambda = \infty$ and reviewers provide total rankings of assigned papers as BRE-adjusted-scores. We provide further details of this reduction and the full algorithmic description of BRE-adjusted-scores (Algorithm 3) in Appendix B.1. This reduction is of interest since it shows that when $\lambda = \infty$, the proposed algorithm still makes reasonable use of the provided rankings to incorporate into quantized scores.

### 4.3.2 Analytical solution when only scores are available

While the solution of Algorithm 1 can be complicated to analyze, it is much easier when we only use scores. Without the ranking information, the only constraints on the output scores are $z_{rp} - 0.5 \leq y_{rp} \leq z_{rp} + 0.5, \forall (r, p) \in \mathcal{A}$. We can then get an analytical solution for Algorithm 1 as below.

**Proposition 2.** *Suppose reviewers only provide quantized scores and no rankings, and each paper receives $\mu$ scores. Then for any reviewer-paper pair $(r, p)$ the scores $\{\widehat{y}_{rp}\}_{(r,p) \in \mathcal{A}}$ output by Algorithm 1 can be written in closed form as*

$$\widetilde{y} = \frac{1 + \mu\lambda}{\mu(1 + \lambda)} z_{rp} + \sum_{r' : (r', p) \in \mathcal{A}, r' \neq r} \frac{1}{\mu(1 + \lambda)} z_{r'p}$$

$$\widehat{y}_{rp} = \begin{cases} \widetilde{y} & \text{if} \quad \widetilde{y} \in [z_{rp} - 0.5, z_{rp} + 0.5] \\ z_{rp} - 0.5 & \text{if} \quad \widetilde{y} < z_{rp} - 0.5 \\ z_{rp} + 0.5 & \text{if} \quad \widetilde{y} > z_{rp} + 0.5. \end{cases}$$

The full proof is provided in Appendix B.2. The output $\{\widehat{y}_{rp}\}_{(r,p) \in \mathcal{A}}$ is a convex combination of scores for paper $p$, where the weights depend on the constant $\mu$ as well as the hyperparameter $\lambda > 0$. Precisely, the relative weight on the quantized score given by reviewer $r$ is $1 + \mu\lambda$ whereas the weights on every other reviewer are 1 (note that the number of reviews per paper $\mu$ is typically a small constant). Assume that the scores of paper $p$ are not the same across all reviewers. Then, as the hyperparameter $\lambda$ increases and the weight on consensus decreases, we can see from the reduction in Proposition 2 that the estimated score $\{\widehat{y}_{rp}\}$ approaches $\{z_{rp}\}$ as one should expect. As a further special case, when the scores received by any

paper $p$ from all reviewers are identical to, say, $z_p$, the estimates $\widehat{y}_{rp}$ also reduce to $z_p$ as one may expect. When $\lambda = \infty$, the output is simply $z_{rp}$. This result proves reasonable behavior from the proposed algorithm when only score information is available.

### 4.3.3 Connection to Thurstone model with quantization when only scores are available

We now draw a connection between the consensus objective used in Algorithm 1 and the Thurstone model with quantization. In this section, we retain the assumption in Proposition 2 that only scores are provided and focus on the consensus objective. The Thurstone model (Thurstone, 1927) is a widely used statistical model. For example, it is used for modeling peer grading in MOOCS in Piech et al. (2013). In the Thurstone model with quantization, scores are generated by the following process. Each paper is assumed to have an underlying true quality score $x_p^*$. The latent evaluation score $y_{rp}$ given by reviewer $r$ to paper $p$ is generated from a normal distribution, whose mean is the underlying true score of paper $p$. That is, $y_{rp} \sim \mathcal{N}(x_p^*, \sigma^2)$, for some value $\sigma$ that represents the standard deviation. Our setting involves quantization, we then assume the observed scores $\{z_{rp}\}_{(r,p)\in\mathcal{A}}$ are quantized scores obtained by quantizing the latent scores $\{y_{rp}\}_{(r,p)\in\mathcal{A}}$ to integers, that is, $z_{rp} = \lfloor y_{rp} \rceil, (r,p) \in \mathcal{A}$. [3] We focus on analysis of variables $\{y_{rp}\}_{(r,p)\in\mathcal{A}}$. Since the quantization process from $y$ to $z$ is itself deterministic, we consider the joint likelihood $\mathbb{P}(z, y; x^*)$. To maximize the likelihood, we take the maximization over both $x^*, y$. We focus on the solutions for $y$, which we aim to connect our output to. The connection between the consensus objective (which uses scores) and the log-likelihood under the Thurstone model with quantization is made in the following proposition.

**Proposition 3.** *The maximizer $\{y_{rp}\}_{(r,p)\in\mathcal{A}}$ of the consensus objective has the largest likelihood under the Thustone model with quantization, that is,*

$$
\underset{\{y_{rp}\}_{(r,p)\in\mathcal{A}}}{\arg\min} \quad \sum_{p\in[P]} \sum_{r:(r,p)\in\mathcal{A}} \left( y_{rp} - \frac{1}{|\{r' : (r',p) \in \mathcal{A}\}|} \sum_{r':(r',p)\in\mathcal{A}} y_{r'p} \right)^2
$$

$$
= \underset{\{y_{rp}\}_{(r,p)\in\mathcal{A}}}{\arg\max} \quad \underset{\{x_p^*\}_{p\in[P]}}{\max} \quad \log \mathbb{P}(\{z_{rp}\}_{(r,p)\in\mathcal{A}}, \{y_{rp}\}_{(r,p)\in\mathcal{A}}; \{x_p^*\}_{p\in[P]})
$$

*such that $z_{rp} - 0.5 \le y_{rp} \le z_{rp} + 0.5 \quad \forall\ (r,p) \in \mathcal{A}$.*

The proof of this proposition is provided in Appendix B.3. We thus see that our consensus principle (Design Principle 3) also follows if we consider a standard parametric model.

### 4.4 Hyperparameter selection via Quantization Validation (QV)

In this section, we introduce a novel cross-validation-like method to choose the appropriate value of hyperparameter $\lambda > 0$ in Algorithm 1. The selected value of $\lambda$ is then given as an input to the algorithm. In a hypothetical situation where we were to observe ground truth values $y_{rp}$ for some pairs of $(r,p) \in \mathcal{A}$, it would be possible to perform traditional cross-validation where we choose the value of $\lambda$ that achieves the best performance in estimation of the observed values of $y_{rp}$ on a holdout set. However, in practice, we only have access to the quantized scores. This motivates us to design a procedure to select $\lambda$, which performs validation on a dataset constructed by further quantization of the observations. We call this procedure Quantization Validation (QV).

We present our QV procedure in Algorithm 2. In words, given a set of possible values for $\lambda$, we first construct a validation set by further coarsening the observed quantized scores using fewer quantization bins. Then we select the best value of $\lambda$ that achieves the lowest loss in the recovery of the original quantized scores, via a function of our choice $\texttt{loss} : \mathbb{R}^{|\mathcal{A}|} \times \mathbb{R}^{|\mathcal{A}|} \to \mathbb{R}$. We use Kendall-tau ranking error as the loss function, which is defined formally in the experiments Section 5.1. Specifically, we select a quantization function $q : \mathbb{R} \to \mathbb{R}$, to convert the quantized score $z_{rp}$ to $z'_{rp}, \forall (r,p) \in \mathcal{A}$ that has fewer quantization levels. For example, in our subsequent experiments, we choose $q(\cdot) = \lceil \frac{\cdot}{2} \rceil$. The new rankings are then re-computed from the quantized scores $\{z_{rp}\}_{(r,p)\in\mathcal{A}}$ as $\{\pi'_r\}_{r\in[R]}$. We validate on the dataset consisting of $\{z'_{rp}\}_{(r,p)\in\mathcal{A}}$ and $\{\pi'_r\}_{r\in[R]}$, where the goal is to recover $\{z_{rp}\}_{(r,p)\in\mathcal{A}}$. Given a pre-specified set $\Lambda$ of candidate values of $\lambda$, we compare the

---

[3]The notation $\lfloor \cdot \rceil$ denotes the mapping from any real value to the nearest integer.

---

**Algorithm 2** Quantization Validation (QV)

---

**Require:** *Inputs*: observations of quantized scores $\{z_{rp}\}_{(r,p)\in\mathcal{A}}$, rankings $\{\pi_r\}_{r\in[R]}$, set of possible values of the hyperparameter $\Lambda$, quantization function $q$, loss function `loss`.
 1: Calculate further-quantized scores $z'_{rp} = q(z_{rp})$, $\forall (r,p) \in \mathcal{A}$.
 2: Calculate rankings
 3:    $\pi'_r = \{(p, p') : z_{rp} > z_{rp'}, (r, p) \in \mathcal{A}, (r, p') \in \mathcal{A}\}$.
 4: **for** $\lambda \in \Lambda$ **do**
 5:    Compute the validation error $e_\lambda = \texttt{loss}(z, \widehat{z})$, where $z$ is the vector of $\{z_{rp}\}_{(r,p)\in\mathcal{A}}$ and $\widehat{z}$ of $\{\widehat{z}_{rp}\}_{(r,p)\in\mathcal{A}}$.
 6: **end for**
 7: *Output* selected value of hyperparameter $\lambda \in \arg\min_{\lambda\in\Lambda} e_\lambda$. (Ties broken in favor of the smallest value.)

---

algorithm output $\{\widehat{z}_{rp}\}_{(r,p)\in\mathcal{A}}$ under each value of $\lambda$ with its ground truth $\{z_{rp}\}_{(r,p)\in\mathcal{A}}$, and compute the ranking error. Finally, we select the value of $\lambda$ as the one that induces the smallest ranking error in the validation process, where ties are broken in favor of the smallest value.

## 5 Experiments

We evaluate the empirical performance of our proposed algorithm on simulated data as well as on a semi-synthetic dataset based on real-world data collected from the peer review process of the ICLR 2017 conference. The code for our algorithms and results is available online. [4]

### 5.1 Implementation details

For each experiment, we run 20 trials and plot the mean and standard error of the mean. We treat differences less than $10^{-4}$ in values of $\widehat{y}_{rp}$ as ties.

**Proposed algorithm.** As discussed in Section 4.2, Algorithm 1 solves a strictly convex optimization problem with linear inequalities. We use CVXPY to obtain the solutions: The solver we used is CVXOPT with the tolerance for feasibility conditions (feastol) set as $10^{-6}$. The constant $\epsilon$ which enforces the strict inequality constraints is set as 0.05. Note that the algorithm is robust to the choice of $\epsilon$ (as long as it is small, such that the problem in Algorithm 1 is feasible): we present experiments demonstrating its robustness to the choice of $\epsilon$ in Appendix A.4.

**Quantization Validation.** We use an exponential grid for candidate values of $\lambda$ in Quantization Validation: $\Lambda = \{\exp(t/4) : 0 \le t < 40, t \in \mathbb{Z}\}$. The quantization function is set to $q(\cdot) = \lceil\frac{\cdot}{2}\rceil$.

**Performance measure.** We use the (normalized) Kendall-tau ranking error as the loss function. Intuitively, given two rankings on the same set of elements, the Kendall-tau ranking error measures the number of pairs of items whose relationships are reversed in the two rankings. More formally, we define $y$ to be the vector of $\{y_{rp}\}_{(r,p)\in\mathcal{A}}$ and $\widehat{y}$ to be the vector of $\{\widehat{y}_{rp}\}_{(r,p)\in\mathcal{A}}$. Then the normalized Kendall-tau ranking error between $y$ and $\widehat{y}$ is defined as the following.

$$\frac{1}{|\{((r,p),(r',p')) : y_{rp} > y_{r'p'}\}|} \sum_{(r,p),(r',p'):y_{rp}>y_{r'p'}} \left( \mathbb{I}(\widehat{y}_{rp} \prec \widehat{y}_{r'p'}) + \frac{1}{2}\mathbb{I}(\widehat{y}_{rp} = \widehat{y}_{r'p'}) \right).$$

Note that the pairs $((r,p),(r',p'))$ which are tied in $y$ are omitted from the computation above. In addition to the ranking error, we also measure the $\ell_2$-error between the output values $\{\widehat{y}_{rp}\}_{(r,p)\in\mathcal{A}}$ and the ground truth values $\{y_{rp}\}_{(r,p)\in\mathcal{A}}$ as $\sqrt{\sum_{r,p:(r,p)\in\mathcal{A}} |y_{r,p} - \widehat{y}_{r,p}|^2}$.

---

[4]https://github.com/MYusha/rankings_and_quantized_scores

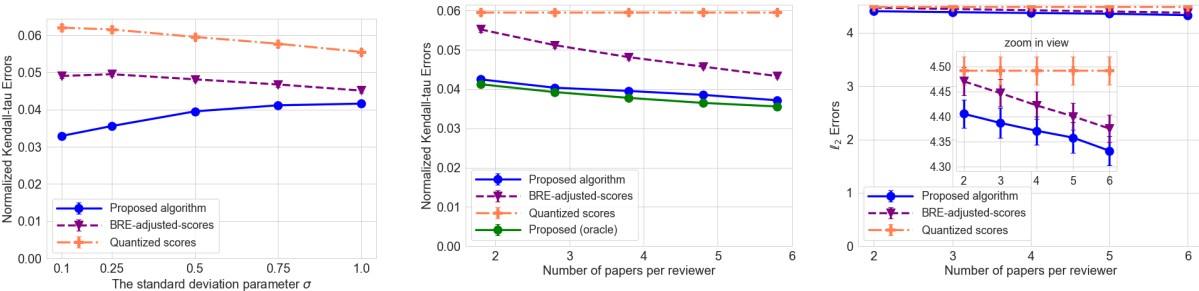

(a) Varying standrad deviation $\sigma$.

(b) Varying number of papers per reviewer.

(c) $\ell_2$-error. Varying number of papers per reviewer.

Figure 2: Experiment results on synthetic data with varying levels of noise (standard deviation $\sigma$) and number of papers assigned to each reviewer.

## 5.2 Baseline methods

We compare the dequantized scores output by our algorithm with the outputs of the following two natural baseline methods.

1. Quantized scores: We simply use the observed scores $\{z_{rp}\}_{(r,p)\in\mathcal{A}}$ as the final output.

2. BRE-adjusted-scores (Algorithm 3): In Section 4.3, we have shown that this algorithm is equivalent to our proposed algorithm when $\lambda \to \infty$ and reviewers provide total rankings of assigned papers. In ICLR 2017 dataset, reviewers may provide partial rankings instead. We make simple adjustments to the baseline to accommodate, which are explained in Appendix C. At a high level, the partial rankings can be viewed as total rankings over (tied) groups of papers. Therefore, the adjustments to the baseline is that the output scores are adjusted from the quantized scores in groups instead of individually. For simplicity, we refer to this baseline in the ICLR 2017 dataset as BRE-adjusted-scores as well, throughout this section.

## 5.3 Synthetic dataset

We evaluate the performance of the proposed algorithm when the data is generated from a Thurstone model. For $P$ papers to evaluate, their true scores are drawn independently from the uniform distribution $x_p^* \sim \text{Unif}[1,9] \; \forall p \in [P]$. The latent unquantized score $y_{rp}$ that is given by reviewer $r$ to paper $p$ is drawn from the normal distribution: $y_{rp} \sim \mathcal{N}(x_p^*, \sigma^2)$, where the parameter $\sigma$ represents the standard deviation of the model, then clipped by $[0,10]$. The quantized scores are then generated by rounding $\{y_{rp}\}_{(r,p)\in\mathcal{A}}$ to the nearest integer such that $0 \le z_{rp} \le 10, z_{rp} \in \mathbb{Z}, \; \forall(r,p) \in \mathcal{A}$. For simplicity, we assign the same number of papers to each reviewer in the experiments. Similarly, we give the same number of reviewers (scores) to each paper. Given fixed numbers of papers per reviewer and reviewers per paper, the reviewer assignment $\mathcal{A} = \{(r,p) \text{ where } r \text{ reviews } p, r \in [R], p \in [P]\}$ is generated uniformly at random from all possible assignments. The default setting is set to $P = 60$, $\sigma = 0.5$, each paper gets 4 reviewers and each reviewer is assigned 4 papers. We examine the performance of our proposed algorithm under various settings where we change the parameters individually.

**Varying levels of noise.** We obtain solutions with data generated from the Thurstone model with different values of the standard deviations and show the results in normalized Kendall-tau error in Figure 2a. Note that under the setting with very small noise, the performances of the two baselines are slightly worse than those under the setting with a larger noise. This is caused by a larger number of ties between the quantized scores under a smaller noise level. Specifically, when $\sigma = 1.0$, the average percentage of ties in the output of the quantized scores and the BRE-adjusted-scores are 11.1% and 5.8% respectively. However, when $\sigma$ decreases to 0.1, the average percentage of ties in their output increases to 12.4% and 6.3%. By considering both

consensus between reviewers and the provided rankings in dequantizing the scores, our proposed algorithm has less than 1.5% ties in its output and outperforms the baseline methods under all three settings. We further show the distribution of selected $\lambda$ over the 20 trials in Figure 6 in Appendix A. As the noise level increases, the distribution of $\lambda$ selected by QV moves toward the larger end, which is an indication of decreasing weight on the consensus term in the objective in Algorithm 1.

**Varying loads.** We also consider settings with varying numbers of assigned papers to each reviewer and varying numbers of reviews for each paper. While varying one of the two variables, we fix the other to be the same as the default setting. The error rates with a varying number of papers assigned to each reviewer are shown in Figure 2b. As the number of assigned papers increases, the error of BRE-adjusted-scores baseline decreases, similar to the proposed algorithm. This is because the simple comparison injection employed by the BRE-adjusted-scores baseline lets the number of papers per reviewer directly dictate the number of possible values for dequantized scores. Results show that changes in the number of reviewers per paper do not affect the performance significantly so we defer the corresponding plot of error rates to Figure 5 in Appendix A. In all settings with varying loads, our proposed algorithm consistently incurs smaller errors than the baselines. We present an additional result (Figure 2b) which is the performance of the proposed algorithm under the oracle $\lambda^*$ ("Proposed (oracle)"). The oracle $\lambda^*$ is the one that achieves the lowest error on the dataset, among the exponential grid for candidate values (Section 5.1). The small gap between the proposed algorithm with QV-selected $\lambda$ and with the oracle $\lambda^*$ demonstrates the strong performance of QV. More results regarding the hyperparameter selection process are in Section 5.5.

**Additional experiment settings.** To further validate the effectiveness of the proposed algorithm, we also present results in the following experiment settings, per the suggestions of reviewers for our paper.

1. **Varying number of papers** (Appendix A.5). In practice, the number of submissions would often be larger, ranging from hundreds to thousands. We present experiments with larger number of papers ranging from 200 to 2000 and observe that the performances remain similar as the number increases.

2. **Varying affinities between reviewers and papers** (Appendix A.6). We consider the more general setting where each latent score $y_{r,p}$ in the Thurstone model has a different variance $\sigma_{r,p}$, to simulate different underlying affinities between papers and reviewers (Stelmakh et al., 2019b). The proposed algorithm retains its advantage in this setting.

$\ell_2$ **error.** In addition to the ranking error, we also report $\ell_2$ errors on the synthetic dataset. The $\ell_2$ errors are calculated with the same set of dequantized scores for which we report the ranking errors. Despite its primary focus on recovering the ranking among $y_{rp(r,p)\in\mathcal{A}}$, the proposed algorithm shows a small advantage in $\ell_2$ error as well, compared to the baselines. The $\ell_2$ errors with varying numbers of papers per reviewer are displayed in Figure 2c. The results on the $\ell_2$ error for varying the noise and loads are qualitatively similar in that our method is no worse and offers a small improvement. The plots are deferred to Figure 7 in Appendix A.3.

### 5.4 Semi-synthetic dataset based on real data from ICLR 2017

We conduct experiments on data from the peer-review process of the ICLR 2017 conference (Kang et al., 2018). There are $P = 427$ papers and every paper receives at least 3 reviews. For simplicity (so that number of papers is a multiple of the reviewer load), we keep $P = 426$ papers and retain 3 reviews for each paper by discarding some reviews (43 out of 1321 reviews are discarded). Since the reviewers are all anonymous in this dataset, we allocate papers to reviewers randomly (sampling from the entire pool of papers without replacement), subject to the constraint that a fixed number of papers are assigned to every reviewer. There are several other datasets presented by Kang et al. (2018), but those datasets are not suitable for our purposes. In Appendix D, we provide explanations on why those datasets are not suitable.

Each review score comprises an integer from 1 to 10, which we treat as the ground-truth values for unquantized scores $\{y_{rp}\}_{(r,p)\in\mathcal{A}}$. We generate the quantized scores by putting the original review scores in

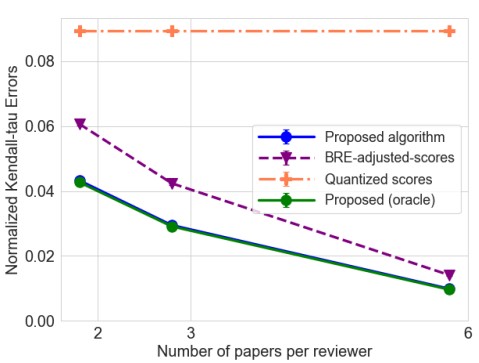
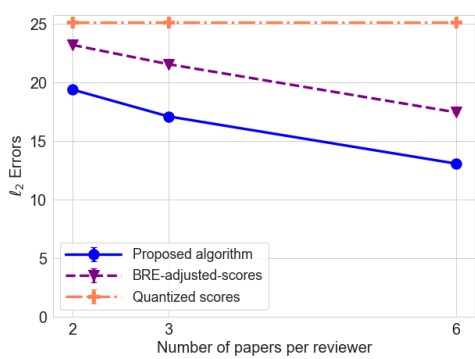

(a) Varying number of papers per reviewers.

(b) $\ell_2$-error. Varying number of papers per reviewers.

Figure 3: Experiment results on ICLR 2017 data.

fewer quantization levels, via the quantization procedure: $z_{rp} = \lceil y_{rp}/2 \rceil$, such that $1 \leq z_{rp} \leq 5, z_{rp} \in \mathbb{Z}, \ \forall (r,p) \in \mathcal{A}$. The reviewer-provided partial rankings are generated from the original review scores as $\pi_r = \{(p,p') : y_{rp} > y_{rp'}, (r,p) \in \mathcal{A}, (r,p') \in \mathcal{A}\}$, since the ICLR 2017 conference did not collect rankings directly from reviewers.

Figure 3a plots the normalized Kendall-tau error of our algorithm and the baselines under varying numbers of papers per reviewer (note that we cannot control the noise level $\sigma$ in the ICLR 2017 dataset as we do in the synthetic dataset.) We observe that our algorithm incurs a lower error as compared to the baselines. Furthermore, the error decreases as the number of papers per reviewer increases. As this number decreases from 6 to 2, the percentage of ties in the output of quantized scores remains at 32.2%, while the percentage in the output of BRE-adjusted-scores increases from 8.0% to 23.2%. Our algorithm outputs less than 3.4% percent of ties in all three settings. Similar to Figure 2b, we plot the performance of the proposed algorithm under oracle $\lambda^*$ in Figure 3a ("Proposed (oracle)"). We observe that the performance of the proposed algorithm with oracle $\lambda$ (green line) have negligible difference with the proposed algorithm with QV-selected $\lambda$ (blue line). This shows the effectiveness of QV.

Other than the random reviewer assignment, we also show results under a "clustered" reviewer assignment, per suggestions of reviewers for our paper. In this setting, each reviewer can only reviewer papers from the same cluster, in other words, subset of all papers. This clustered structure simulates different genres of paper, since the real assignment data is not public. Details of these experiments are in Appendix A.7.

Similar to the synthetic dataset, we report the $\ell_2$ errors on the ICLR 2017 dataset in Figure 3b. Again, we observe that our estimator slightly outperforms the baselines in terms of the $\ell_2$ error. More details can be found in Appendix A.3.

## 5.5 Hyperparameter $\lambda$ selection via QV

We present results that shed light on the Quantization Validation (QV) hyperparameter selection process introduced in Section 4.4, as well as the effect of hyperparameter $\lambda$ on the performance of the proposed algorithm. Figure 4 presents plots comparing the performance of different values of hyperparameter $\lambda$, the value chosen by QV, and the best value of $\lambda$ chosen by a hypothetical oracle that has access to ground truth data. The plots compare these choices on two synthetic datasets (with parameters set as defaults specified in Section 5.3) and the ICLR 2017 dataset (with 6 reviewers per paper). The errors are shown in log-scale for clarity for ICLR 2017 data, note that y-axes may not start at 0 to be able to zoom in on the relevant parts. Our experiments (see also Figure 2b and Figure 3a) reveal a strong performance of the Quantization Validation process: we observe that it can select a good $\widehat{\lambda}$ close to the optimal ideal value $\lambda^*$, and incurs an error close to that incurred by $\lambda^*$.

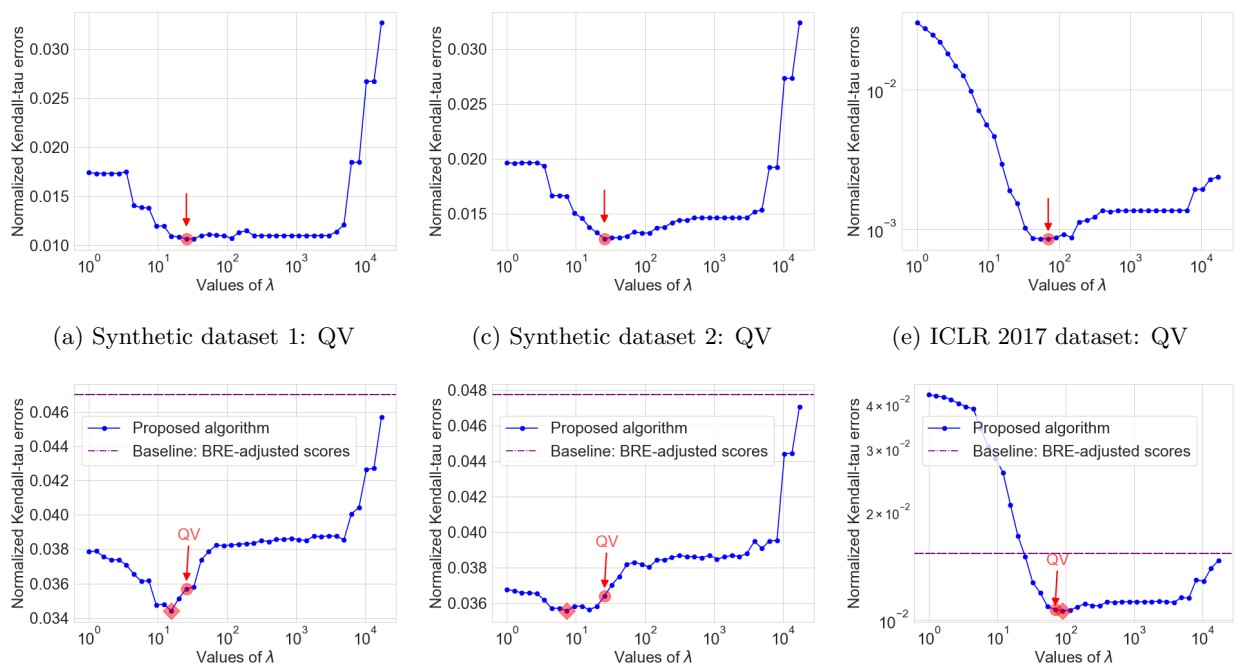

(a) Synthetic dataset 1: QV      (c) Synthetic dataset 2: QV      (e) ICLR 2017 dataset: QV

(b) Synthetic dataset 1: original data    (d) Synthetic dataset 2: original data    (f) ICLR 2017 dataset: original data

Figure 4: The ranking error with varying $\lambda$ in synthetic and ICLR datasets. **Top row:** Error on the quantization validation set. Red circles mark the value selected by QV which achieves the smallest errors on the quantization validation set. **Bottom row:** Error curves on the original dataset. The circles indicate performance under the hyperparameter value selected by QV (from the top row). This is compared to the performance of optimal value marked by red diamonds, which achieves the smallest error on the original dataset. Note that the y-axis does not start at zero, in order to clearly depict the relative error of various hyperparameter choices.

## 6 Discussion

We address the problem of aggregating quantized scores and rankings evaluations in the form of dequantized scores, applicable to important settings such as peer-review. We propose a computationally-efficient algorithm that solves a constrained convex optimization problem to compute the dequantized scores. The proposed algorithm is evaluated on both synthetic data and datasets based on real peer-review scores, and shows strong empirical advantages.

An aspect to keep in mind regarding any such adjustment that uses global data is that of privacy in peer review (Ding et al., 2020; Jecmen et al., 2020; Ding et al., 2022). By providing Area Chairs with the information aggregated across all the reviewers who reviewed their assigned papers, we need to ensure that it should not inadvertently reveal the review information of paper(s) that they are not handling. As illustrated in Figure 1a, the Area Chairs only observe rankings between papers that they are handling. For example, an AC might see "$A > \star > B$" where "$\star$" is paper C whose information is hidden from the AC. In our algorithm, however, the dequantized scores of A and B might depend on the score of paper C (and scores given by other reviewers who reviewed paper C, etc.). If A gets a dequantized score of 6 while B gets 3.95, C might have a tie with B with dequantized score of 4.05. Therefore, the dequantized scores may contain more information than the quantized scores plus the partially-obscured rankings revealed to the AC. Although it is unlikely that the AC would be able to directly infer, for example, the identity of reviewers of paper C, it is a potential concern that some information about paper C would be revealed from the dequantized scores given to the AC. Another direction of future work is that of global versus subgroup accuracy. For example, some subgroups of papers might have a larger inter-reviewer disagreement, or fewer

reviewer-provided comparisons than other papers, because of their fields. This phenomenon might affect the ranking errors in these subgroups through the consensus objective in our algorithm. Therefore, it is also of interest to analyze and/or modify our proposed algorithm to ensure comparable error rates across various subgroups of papers, in addition to the global Kendall-tau ranking error that we currently study in this work. Finally, it is of interest to prove strong theoretical guarantees about the proposed algorithm including our Quantization Validation method, or design new algorithms for this problem that have strong theoretical guarantees with good empirical performance.

## 7 Acknowledgements

This work was supported in parts by NSF CAREER award 1942124, NSF CIF 1763734, and a Google Research Scholar Award. We thank the AE Branislav Kveton and the anonymous reviewers for their valuable and prompt comments and suggestions on the manuscript.

**Broader Impact Statement**

The interest for an efficient and accurate peer review system has dramatically grown for modern large scientific conferences. This work presents a practical attempt at using pairwise comparisons to improve the evaluation scores in conference peer-review and potentially, in other settings where quantized scores and comparison information are both available. As outlined in Section 6, practitioners should also be aware of the potential negative effects of our proposed method before applying it to a real setting, including the potential reveal of confidential review information and possible worse accuracy on certain subgroups of papers/reviewers. The quality of the updated scores from our algorithm might be affected by other problems in peer-review, such as malicious behaviors from paper authors. For example, a reviewer who is also a paper author might intentionally give low quantized scores or manipulate the rankings of assigned papers to improve the acceptance chance of their own paper. See Section 2 for references to works that identify and propose solutions for those (parallel) problems, although it remains to connect them with our problem formulation. Evaluating these aspects forms useful directions for future work.

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

# A Additional experimental results

In this appendix, we present additional experimental results supplementing those in Section 5.

## A.1 Performance with varying numbers of reviewers per paper

In Figure 5, we show ranking error rates on the synthetic dataset with varying numbers of reviewers per paper. Change of this number does not affect the algorithm's performance significantly.

## A.2 Distribution of hyperparameters selected by QV

In Figure 6, we show the distribution of hyperparameter value chosen by the QV procedure (Algorithm 2) on synthetic datasets. The distribution of values changes most significantly as the noise standard deviation $\sigma$ increases. This is because increasing $\sigma$ makes scores more divergent for each paper. The QV effectively captures this and as a result, chooses larger $\lambda$ as the noise level $\sigma$ increases, which leads to decreasing weight on the consensus term in the objective of the optimization problem in Algorithm 1. The change in the number of papers per reviewer, or the number of reviewers per paper does not significantly affect the range of hyperparameter values selected by the QV. This indicates that these two parameters have no significant effect on how much the algorithm relies on consensus, as opposed to the noise level $\sigma$.

## A.3 Performance in $\ell_2$ errors

We display the additional results in $\ell_2$ error on synthetic data in Figure 7. For the ICLR 2017 dataset, where the $\ell_2$ errors are shown in Figure 3b, we provide additional details as follows. Given the data-generation process, we first project the dequantized scores back to integers in $[1, 10]$ by the function $\widehat{y}_{r,p} = \lfloor 2\widehat{y}_{r,p} - 0.5 \rfloor, (r, p) \in \mathcal{A}$. For example, $\widehat{y}_{rp}$ in the interval $[1, 1.5)$ is projected to 2, and $\widehat{y}_{rp}$ in the interval $[1.5, 2)$ is projected to 3. For synthetic dataset, we provide the additional results of performance in $\ell_2$ error with varying levels of noise $\sigma$ and numbers of reviews per paper. The results under these settings are shown in Figure 7.

## A.4 Performance with varying $\epsilon$

We show that the proposed algorithm is robust to several choices of small $\epsilon$. We vary the value of $\epsilon$ for our proposed algorithm and obtained results on both synthetic and ICLR 2017 data, which are shown in Figure 8. For synthetic data, the parameters are set to default (Section 5.3). For the ICLR 2017 data, the only parameter which is the number of papers per reviewer is set to 6. Performances of the baselines are also plotted as a reference.

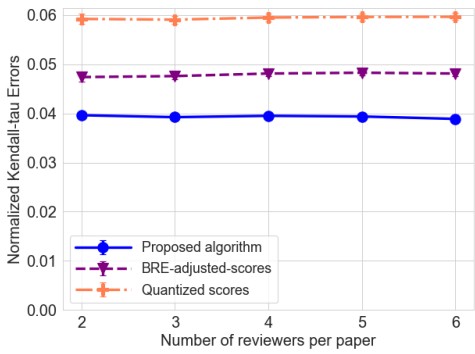

Figure 5: Synthetic data with varying numbers of reviewers per paper.

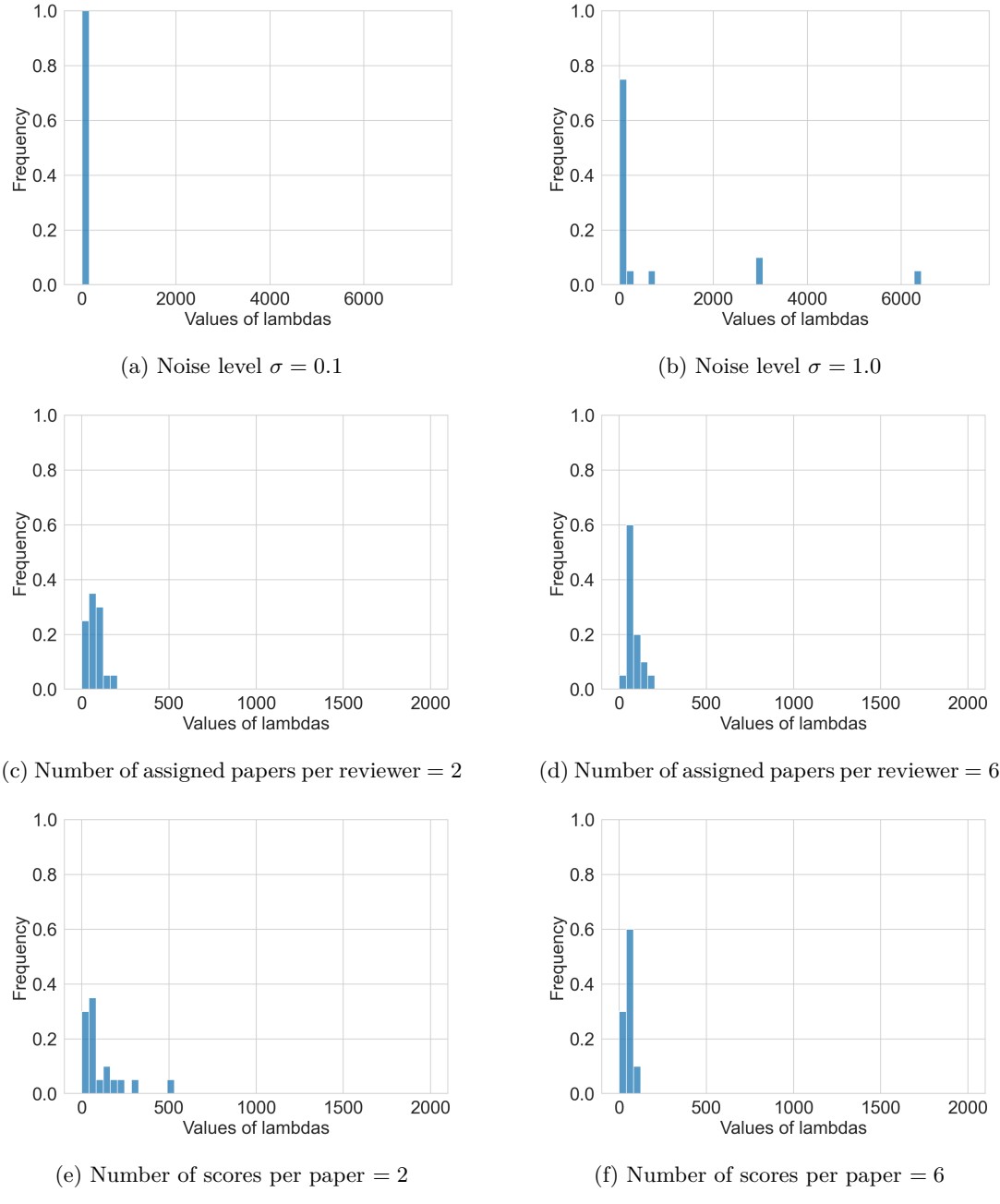

(a) Noise level $\sigma = 0.1$

(b) Noise level $\sigma = 1.0$

(c) Number of assigned papers per reviewer $= 2$

(d) Number of assigned papers per reviewer $= 6$

(e) Number of scores per paper $= 2$

(f) Number of scores per paper $= 6$

Figure 6: Distributions of $\lambda$ selected by QV in synthetic dataset with varying parameters.

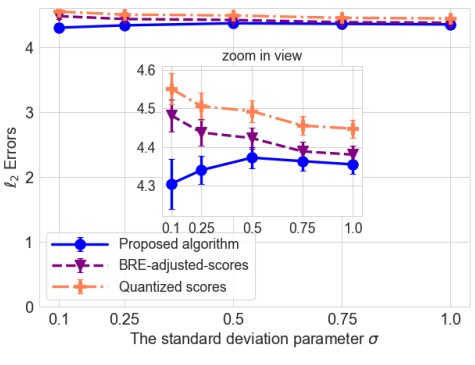

(a) Varying noise level $\sigma$.

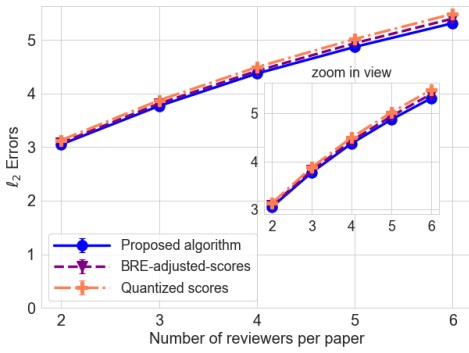

(b) Varying number of reviewers per paper.

Figure 7: Additional experiment results on $\ell_2$ errors on synthetic data.

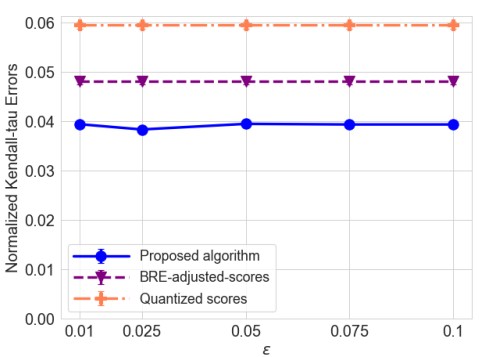

(a) Experimental results on synthetic data.

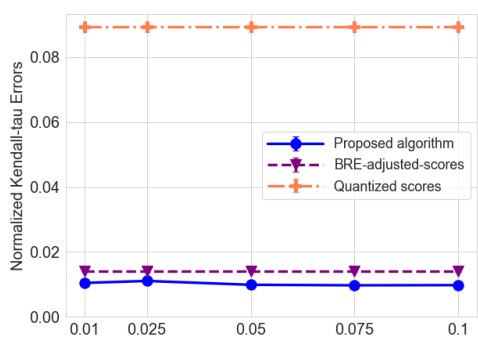

(b) Experimental results on ICLR 2017 data.

Figure 8: Experimental results with varying $\epsilon$.

## A.5 Performance with large number of papers

To validate the performance of the proposed algorithm on datasets with large numbers of papers, we conduct additional experiments with synthetic data, increasing the total amount of papers to 2000. The other parameters are set to default (Section 5.3). The results are plotted in Figure 9. Similar to the numbers of reviewer per paper (Figure 5]), the total number of papers does not have a significant impact on the

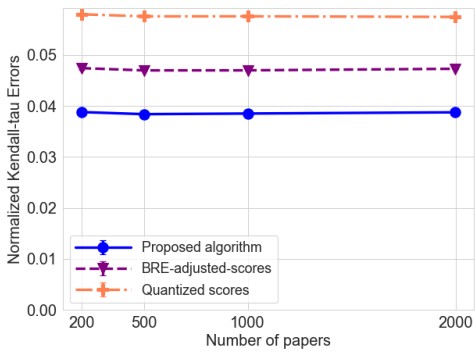

Figure 9: Experiment results on Synthetic data with large number of papers.

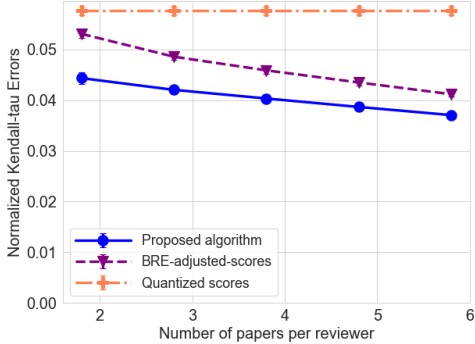

Figure 10: Experiment results on Synthetic data with variances from uniform distribution.

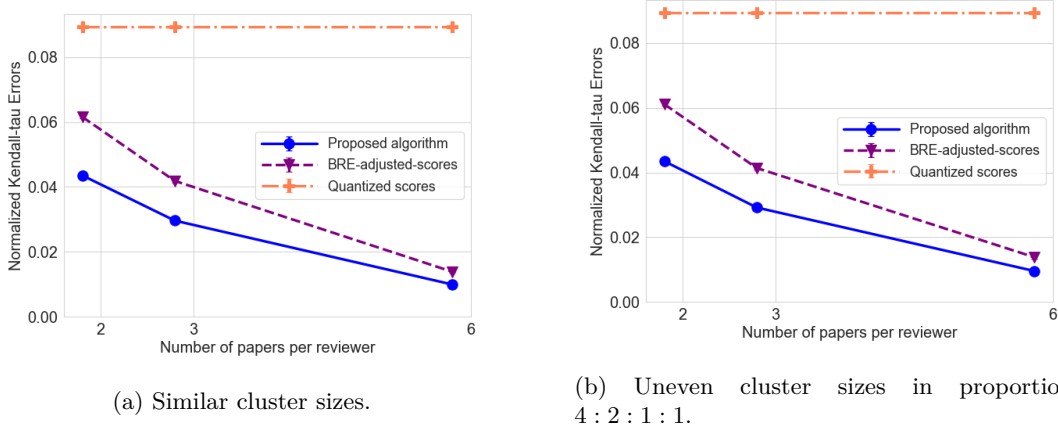

(a) Similar cluster sizes.

(b) Uneven cluster sizes in proportion $4 : 2 : 1 : 1$.

Figure 11: Experiment results on ICLR 2017 data with clustered reviewer assignment.

normalized Kendall-tau ranking error (Section 5.1). The results show that the proposed algorithm continues to outperforms baselines as the total number of papers grows large.

### A.6   Performance with varying affinities between reviewers and papers

In the Thurstone model (Section 4.3.3), the variance $\sigma^2$ in the normal distribution of the latent score $y_{r,p}$ can be viewed as a presentation of the underlying affinity between paper $p$ and reviewer $r$ (Stelmakh et al., 2019b), in other words, the reviewer's expertise on their assigned paper. In our synthetic experiments (Section 5.3), the variance $\sigma^2$ is set to be the same for all reviewer-paper pairs. In this subsection, we let $y_{r,p} \sim \mathcal{N}(x_p^*, \sigma_{r,p}^2)$, for each $(r,p)$ in the assignment $\mathcal{A}$, $\sigma_{r,p}$ is drawn independently and identically from a uniform distribution $U[0.1, 0.9]$. We plot the results under several choices of numbers of assigned papers per reviewer in Figure 10. Our proposed algorithm remains dominant in the performance in various settings.

### A.7   Performance with clustered reviewer assignment

The reviewer assignment in the ICLR 2017 dataset is generated randomly (Section 5.4) because the real reviewer assignment data is not public. In this subsection, we experiment with the following "clustered" reviewer assignments as a middle ground. Note that since the review scores in ICLR 2017 dataset are given, we cannot vary the reviewer-paper affinities in the same way as in the Thurstone model in Appendix A.6. We divide the papers in the ICLR 2017 dataset randomly into several clusters, and a reviewer can only review papers in the same cluster. The clustered structure simulates different genres of papers, for example, the Theory of ML, Deep learning, etc. We choose the number of clusters to be 4 and considered two options for

cluster sizes: (i) all clusters are of approximately the same sizes and (ii) cluster sizes are uneven. Within each cluster, we assign papers to reviewers randomly. We plot the results for these two cluster sizes in Figure 11. We observe that under the implemented clustered assignments, the performances remains similar to those in Figure 3a.

# B    Proofs of Propositions

In this section, we present the proofs for propositions from Section 4.3.

## B.1    Proof of Proposition 1

For clarity, we first present the full procedure of BRE-adjusted-scores here. It is also one of the baseline methods that we evaluate in Section 5, for synthetic data and with a simple adjustment for the ICLR 2017 data. For conciseness, we fix the number of assigned papers across reviewers and denote this constant as $\kappa$.

---

**Algorithm 3** BRE-adjusted-scores

---

**Require:** The ranking and quantized scores provided by each reviewer: $\{z_{rp}\}_{(r,p)\in\mathcal{A}}$ and $\{\pi_r\}_{r\in[R]}$. For simplicity, assume $\pi_r$ is total ranking among papers $\pi_r = \{p_\kappa \succ p_{\kappa-1} \succ \cdots \succ p_1\}$. A small constant $\epsilon$.

1:  **for** $r \in \{\text{reviewers}\}$ **do**
2:      Divide all the reviewed papers by their quantized scores, denote the set of quantization bins as $\mathbb{B}$.
3:      **for** $\mathcal{B} \in \mathbb{B}$ **do**
4:          Define the ranked papers in $\mathcal{B}$ as $\{p_u \succ p_{u-1} \succ \cdots \succ p_v\}$, where $u \geq v$ and $u - v + 1$ represents number of papers in this quantization bin. Denote the score value which is the same for all papers in $\mathcal{B}$ as $z_{\mathcal{B}}$.
5:          **for** $t = v \ldots u$ **do**
6:              Set $\widehat{y}_{r,p_t} = z_{\mathcal{B}} + (t - v) \times \epsilon$.
7:          **end for**
8:          Adjust the values $\widehat{y}_{r,p} = \widehat{y}_{r,p} - \left( \frac{1}{|\{p:p\in\mathcal{B}\}|} \sum_{p:p\in\mathcal{B}} \widehat{y}_{r,p} - z_{\mathcal{B}} \right)$ for $p \in \mathcal{B}$.
9:      **end for**
10: **end for**
11: Output $\{\widehat{y}_{rp}\}_{(r,p)\in\mathcal{A}}$

---

We first introduce the notion of quantization bins, which is defined separately across different reviewers. For a reviewer $r$, a quantization bin is a group of papers with the same quantized scores given by $r$. Each reviewer can have at least 1 quantization bin and at most $\kappa$ quantization bins in the assigned papers. In Algorithm 3, the quantized score $z_{rp}$ is incremented by a value according to the rank of $p$ inside its quantization bin, then "centered" to form the output whose mean value inside each quantization bin remains the same. Precisely, $\widehat{y}_{rp}$ can be explicitly written as follows. Note that in Algorithm 3, the choice of $\epsilon$ does *not* change the ranking of output $\{\widehat{y}_{rp}\}_{(r,p)\in\mathcal{A}}$ and for implementation, we choose $\epsilon$ to be the same as in the proposed algorithm, which is 0.05.

$$\widehat{y}_{rp} = z_{rp} + \epsilon \left( \pi_r(p) - \frac{1}{2} - \frac{P_{rp}}{2} \right), \tag{1}$$

where $P_{rp}$ denotes the total number of papers in the bin that $p$ belongs to, within the scope of reviewer $r$, and $\pi_r(p)$ denotes the ranking of $p$ inside its quantization bin and within the scope of papers reviewed by $r$. For example, if reviewer $r$ reviews only two papers $p$ and $p'$, and gives identical scores to the two papers, and ranks them as $p \succ p'$, then we have $\pi_r(p) = 2, \pi_r(p') = 1$.

Having established the procedure in Algorithm 3, we now study (i) the reduction from our proposed algorithm as $\lambda = \infty$ to BRE-adjusted-scores, and (ii) the connection between BRE-adjusted-scores and BRE.

### B.1.1 Reduction from proposed algorithm to BRE-adjusted-scores

Recall that, as $\lambda = \infty$, the proposed algorithm reduces to the following optimization problem which does not have the objective term that captures reviewer consensus. $\widehat{y}_r$ is the vector of $\{\widehat{y}_{r,p}\}_{(r,p)\in\mathcal{A}}$ and $y_r, z_r$ are that of $\{y_{r,p}\}_{(r,p)\in\mathcal{A}}, \{z_{r,p}\}_{(r,p)\in\mathcal{A}}$.

$$\widehat{y}_r = \arg\min_{y_r \in \mathbb{R}^\kappa} \|y_r - z_r\|_2, \tag{2}$$

such that $y_{rp} \geq y_{rp'} + \epsilon$ whenever $(r,p) \in \mathcal{A}, (r,p') \in \mathcal{A}$, and $p \succ_r p'$

$$z_{rp} - 0.5 \leq y_{rp} \leq z_{rp} + 0.5 \quad \forall\, (r,p) \in \mathcal{A}. \tag{3}$$

Recall that we assume $\epsilon$ to be a small constant such that the feasible set is not empty. For example, in practice we choose $\epsilon$ to be 0.05 (Section 5). Given the problem in (2), the solutions $\widehat{y}_{rp}$ are different from $z_{rp}$ only when there exist tied scores given by a reviewer. Precisely, with the assumption that reviewers report total rankings, consider the problem of finding the vector $\mathbf{x} = [x_1 \dots x_m] \in \mathbb{R}^m$ such that $x_1 = x_2 + \epsilon = \cdots = x_m + (m-1)\epsilon$ that minimizes the objective function $\|\mathbf{x} - \mathbf{c}\|_2^2$, $\mathbf{c} = [c, c, \dots, c] \in \mathbb{R}^m$ where $c$ is some constant. Let us set the derivative of the objective function with respect to $x_m$ to find the minimizer:

$$\widehat{x}_m = \arg\min_x \sum_{i=0}^{m-1} (x + i\epsilon - c)^2 \quad \rightarrow \quad \widehat{x}_m = c - \frac{m-1}{2}\epsilon.$$

Therefore $\widehat{x}_i = \widehat{x}_m + (m-i)\epsilon = c + \left((m-i) - \frac{m-1}{2}\right)\epsilon$. In the setting of (2), $\mathbf{x}$ corresponds to the vector of scores of papers with tied scores for a fixed reviewer. $c$ is the score value, $m$ is the total number of papers in the quantization bin. Therefore, we have the solution to the problem in (2) as:

$$\widehat{y}_{rp} = z_{rp} + \epsilon\left((m-i) - \frac{m-1}{2}\right) = z_{rp} + \epsilon\left(\pi_r(p) - 1 - \frac{P_{rp}-1}{2}\right)$$

$$= z_{rp} + \epsilon\left(\pi_r(p) - \frac{1}{2} - \frac{P_{rp}}{2}\right). \tag{4}$$

It is easy to see the equivalence between (1) and (4), indicating that under the assumption that each reviewer provides total ranking $\pi_r$ among all assigned papers, the proposed algorithm reduces to Algorithm 3 as $\lambda = \infty$.

### B.1.2 Relationship between BRE-adjusted-scores and BRE

Given possibly noisy, randomly collected pairwise comparisons where each pair is compared with a certain probability, the BRE estimates a score for each item (paper). The final goal in their work is global ranking, which is revealed by the estimated scores. In their setting, each pair can only be compared once, and the score of an item is calculated as the relative number of items preceding and succeeding it. Using our notations, let us denote the output score of paper $p$ as $\widehat{x^*}_p$, since they do not distinguish between different reviewers. In BRE algorithm, the estimated scores for a paper is

$$\widehat{x^*}_p \propto |p' \in [P] : p' \neq p, p' \prec p| - |p' \in [P] : p' \neq p, p' \succ p|. \tag{5}$$

Recall the adjustment to scores performed by BRE-adjusted-scores in (1). For a pair of reviewer and paper $(r,p)$, let $\mathcal{B}_r(p)$ denote the set of papers reviewed by $r$ and are in the same quantization bin as $p$. We can write that:

$$P_{rp} = |p' : p \in \mathcal{B}_r(p), p' \prec p| + |p' : p' \in \mathcal{B}_r(p), p' \succ p| + 1,$$
$$\pi_r(p) = |p' : p \in \mathcal{B}_r(p), p' \prec p| + 1.$$

Plugging the above back in (1) gives us:

$$\widehat{y}_{rp} = z_{rp} + \frac{\epsilon}{2}\left(|p' : p \in \mathcal{B}_r(p), p' \prec p| - |p' : p \in \mathcal{B}_r(p), p' \succ p|\right) \tag{6}$$

It is easy to see that the adjustment amount conditioned on the quantized score in (6) is proportional to the relative difference between the number of papers preceding and succeeding paper $p$ in its quantization bin. The multiplicative factor is simply the small constant $\epsilon/2$, since the output scores are arbitrary-scale (Section 4.2).

Combining B.1.1 and B.1.2, we proved the following in the special case where (i) reviewers report total rankings of assigned paper and (ii) $\lambda = \infty$ in Algorithm 1: Algorithm 1 adds to each quantized score a value that is proportional to the estimated score from BRE, when BRE is given the ranking information within the quantization bin.

## B.2   Proof of Proposition 2

Recall that $\mu$ is the number of scores received by each paper. Without ranking information, $\{\widehat{y}_{rp}\}$ can be solved separately for each paper $p \in [P]$. For a fixed $p$, the proposed algorithm reduces to the following optimization problem.

$$\underset{\{y_{rp}\}_{(r,p)\in E}}{\arg\min} \sum_{\text{reviewers } r:(r,p)\in \mathcal{A}} (y_{rp} - \bar{y}_p)^2 \quad + \quad \lambda \sum_{(r,p)\in \mathcal{A}} (y_{rp} - z_{rp})^2. \tag{7}$$

$$\text{such that } z_{rp} - 0.5 \le y_{rp} \le z_{rp} + 0.5 \quad \forall \, (r,p) \in \mathcal{A}.$$

When $\lambda > 0$, the objective is a strictly multivariate convex function. For every reviewer $r$ that reviews paper $p$, the partial derivative of objective in (7) is as follows. For simplicity, let us denote the objective in (7) as $f_p$.

$$\frac{\partial f_p}{\partial y_{rp}} = 2 \left( y_{rp} - \frac{1}{\mu} \sum_{r:(r,p)\in \mathcal{A}} y_{rp} + \lambda(y_{rp} - z_{rp}) \right).$$

When $y_{rp} = \frac{1+\mu\lambda}{\mu(1+\lambda)} z_{rp} + \sum_{r'\neq r} \frac{1}{\mu(1+\lambda)} z_{r'p}$, the convex multivariate objective function $f_p$ achieves local minimum since its derivatives achieve 0 for all variables. Since the function is strictly convex, the local minimum is the global minimum. If the global minimum is inside the feasible set defined by the linear inequalities, then the solution to the problem in (7) is

$$\widehat{y}_{rp} = \frac{1+\mu\lambda}{\mu(1+\lambda)} z_{rp} + \sum_{r'\neq r} \frac{1}{\mu(1+\lambda)} z_{r'p}. \tag{8}$$

In other words, $\{\widehat{y}_{rp}\}$ is the weighted average of scores for paper $p$, where the weights are dependent on constant $\mu$ (number of reviews for each paper) and the hyperparameter $\lambda$. If the global minimum is outside the feasible set, observe that the objective function in (7) is convex to each variable $y_{rp}$ if other variables are fixed. Therefore, in this case, the solution $\{\widehat{y}_{rp}\}$ is the closest point in $[z_{rp} - 0.5, z_{rp} + 0.5]$ to the right-hand side in (8). To summarize, the solution to (7) is as follows.

$$\widehat{y}_{rp} = \min \left( \max \left( \frac{1+\mu\lambda}{\mu(1+\lambda)} z_{rp} + \sum_{r'\neq r} \frac{1}{\mu(1+\lambda)} z_{r'p}, \ z_{rp} - 0.5 \right), \ z_{rp} + 0.5 \right). \tag{9}$$

## B.3   Proof of Proposition 3

The likelihood for all $r, p : (r, p) \in \mathcal{A}$ which we study can be expressed as

$$\mathbb{P}\left(\{z_{rp}\}, \{y_{rp}\} | \{x_p^*\}\right) \propto \mathbb{P}\left(\{z_{rp}\} \mid \{y_{rp}\}, \{x_p^*\}\right) \mathbb{P}\left(\{y_{rp}\} \mid \{x_p^*\}\right)$$

We thus have the following equivalence for taking maximization over the latent $y$s and $x^*$s. Note that we study the dequantized scores, so focusing on the solutions for $y_{r,p}$.

$$\underset{\{y_{rp}\}}{\arg\max} \ \underset{\{x_p^*\}}{\max} \mathbb{P}(\{z_{rp}\}, \{y_{rp}\} \mid x_p^*) = \underset{\{y_{rp}\}}{\arg\max} \ \underset{\{x_p^*\}}{\max} \mathbb{P}(\{z_{rp}\} \mid \{y_{rp}\}, x_p^*) \ \mathbb{P}(\{y_{rp}\} \mid x_p^*) \tag{10}$$

Given observations of $\{z_{rp}\}_{(r,p)\in\mathcal{A}}$, the likelihood $\mathbb{P}\left(\{z_{rp}\}_{(r,p)\in\mathcal{A}} \mid \{y_{rp}\}_{(r,p)\in\mathcal{A}}, \{x_p^*\}_{p\in[P]}\right)$ equals 1 only when such consistency is satisfied: $y_{rp} \in [z_{rp} - 0.5, z_{rp} + 0.5], \forall(r,p) \in \mathcal{A}$ and equals 0 otherwise, due to the deterministic nature of the quantization process. If consistency is satisfied, we then consider optimization of the second likelihood on the right-hand side by taking the logarithm of it.

$$\log \mathbb{P}(\{y_{rp}\} \mid x_p^*) = \sum_{r:(r,p)\in\mathcal{A}} -\frac{(y_{rp} - x_p^*)^2}{2\sigma^2} \log(\frac{1}{\sigma\sqrt{2\pi}})$$
$$\propto - \sum_{r:(r,p)\in\mathcal{A}} (y_{rp} - x_p^*)^2. \tag{11}$$

For a fixed paper $p$, we can find the maximizer $\widehat{x^*}_p$ of (11) by setting the derivative to 0, which yields $\widehat{x^*}_p = \frac{1}{|\{r':(r',p)\in\mathcal{A}\}|} \sum_{r:(r,p)\in\mathcal{A}} y_{rp}$ for all $p \in [P]$. Plugging $\widehat{x^*}_p, \forall p \in [P]$ back into (10) and we have that:

$$\underset{\{y_{rp}\}}{\arg\max} \; \underset{\{x_p^*\}}{\max} \log \mathbb{P}(\{z_{rp}\}, \{y_{rp}\}; x_p^*) = \underset{\{y_{rp}\}}{\arg\min} \sum_r (y_{rp} - \frac{1}{|\{r' : (r',p) \in \mathcal{A}\}|} \sum_{r:(r,p)\in\mathcal{A}} y_{rp})^2.$$
$$\text{s.t. } z_{rp} - 0.5 \le y_{rp} \le z_{rp} + 0.5$$

Therefore, the maximizers $\{y_{rp}\}_{r,p}$ of likelihood function under the Thurstone model with quantization is the same as the maximizer of the consensus objective in Algorithm 1, with the constraints that $y_{rp}$'s are consistent with the quantized scores $z_{rp}$'s.

## C   Baseline method for ICLR 2017 data

In Algorithm 3 in Section B.1, we show the baseline method when reviewers provide total rankings of assigned papers. However, in the ICLR dataset (Section 5.4), we derive the reviewer-reported rankings from their original review scores, since ranking information was not collected directly from reviewers. The original scores are integers in $1 \sim 10$, and the rankings are derived as: $\pi_r = \{(p, p') : y_{rp} > y_{rp'}, (r, p) \in \mathcal{A}, (r, p') \in \mathcal{A}\}$. Therefore, the reviewer-reported rankings are not total rankings over assigned papers, whenever there exist ties in the original review scores $\{y_{rp}\}_{(r,p)\in\mathcal{A}}$. Instead, they can be seen as a total ranking over groups of paper. Precisely, the rankings observed in this dataset are a subset of partial rankings that can be expressed as $\pi_r = \{\mathcal{G}_{\mathcal{T}} \succ \mathcal{G}_{\mathcal{T}-1} \succ \ldots \mathcal{G}_1\}$, where each $\mathcal{G}_i, i \in [\mathcal{T}]$ represents a group of paper with tied scores, $p \succ p'$ if $p \in \mathcal{G}_u, p' \in \mathcal{G}_v$ and $u, v : u > v, u \in [\mathcal{T}], v \in [\mathcal{T}]$. For input to algorithms, if $y_{rp} = y_{rp'}$, then $p, p' \in \mathcal{G}_i$ for some $i \in [\mathcal{T}]$.

For the ICLR 2017 dataset with partial rankings, we employ a baseline algorithm that can be seen as a generalization of Algorithm 3. The baseline method for the ICLR 2017 dataset is defined in Algorithm 4. The difference is that output scores are now adjusted from the quantized scores in groups instead of individually. Precisely, the score adjustment of an item is not dependent on the number of items preceding and succeeding it, but on the number of groups preceding and succeeding its group. When reviewers provide total rankings of assigned papers, each group only contains one paper and consequently, Algorithm 4 reduces to Algorithm 3.

In Algorithm 4, the value of $\epsilon$ remains 0.05.

## D   Limitations of the other datasets in Kang et al. (2018)

Kang et al. (2018) present several other datasets besides the one collected from the ICLR 2017 conference. These datasets are 1. arXiv 2007-2017; 2. NeurIPS 2013–2017; 3. ACL 2017; 4. CoNLL 2016. Below, we clarify one by one why these datasets are not suitable for the purposed of this paper.

**arXiv**   There are no review scores available for arXiv submissions.

**NeurIPS**   Only review texts and the confidence scores for accepted papers are available. Furthermore, no review scores are available.

---

**Algorithm 4** Partial-rankings-adjusted-scores

---

**Require:** The ranking and scores provided by each reviewer: $\{z_{rp}\}_{(r,p)\in\mathcal{A}}$ and $\{\pi_r\}_{r\in[R]}$. $\pi_r$ is partial ranking among papers $\pi_r = \{\mathcal{G}_\mathcal{T} \succ \mathcal{G}_{\mathcal{T}-1} \succ \ldots \mathcal{G}_1\}$. A small constant $\epsilon$.

1: **for** $r \in \{\text{reviewers}\}$ **do**

2:     Divide all the reviewed papers by their quantized scores, denote the set of quantization bins as $\mathbb{B}$.

3:     **for** $\mathcal{B} \in \mathbb{B}$ **do**

4:         **if** papers $\{p\}_{p\in\mathcal{B}}$ belong to more than one group **then**

5:             Define the set of groups in $\mathcal{B}$ as $\{\mathcal{G}_u \succ \mathcal{G}_{u-1} \succ \cdots \succ \mathcal{G}_v\}$, where $u \geq v$ and $u-v+1$ represents the number of groups in this quantization bin. Denote the score value for bin $\mathcal{B}$ as $z_\mathcal{B}$.

6:             **for** $t = v \ldots u$ **do**

7:                 Set $\widehat{y}_{r,p} = z_\mathcal{B} + (t-v) \times \epsilon$, for all $p \in \mathcal{G}_t$.

8:             **end for**

9:             Adjust the values $\widehat{y}_{r,p} = \widehat{y}_{r,p} - \left(\frac{1}{|\{p:p\in\mathcal{B}\}|}\sum_{p:p\in\mathcal{B}}\widehat{y}_{r,p} - z_\mathcal{B}\right)$ for $p \in \mathcal{B}$.

10:         **end if**

11:     **end for**

12: **end for**

13: Output $\{\widehat{y}_{rp}\}_{(r,p)\in\mathcal{A}}$

---

**ACL and CoNLL** The original scores in these two datasets are on a scale of 1 to 5. The proposed algorithm (Algorithm 1) runs on data with 5 levels of quantization and works well on such data, as supported by our results on the ICLR 2017 data in which the observed scores have only 5 levels. The only caveat is that, if one additionally wants to evaluate the algorithm's performance by using the original scores as ground truth as we did on the ICLR 2017 data, they would not be able to do so on the ACL or the CoNLL data, due to the small number of levels in their original scores.

