# OpenReview forum: "Integrating Rankings into Quantized Scores in Peer Review"
_TMLR — Accepted by TMLR_

### Review · Reviewer_J9y1 · 2022-05-19

**Summary Of Contributions:**

Thanks to the authors for the hard work on this paper.

The authors address the problem of aggregating quantized scores when ranking are available for the purpose of having higher-quality peer review scores, and (as a result) fewer ties. They use a convex optimization procedure to combine quantized scores and partial rankings into estimated dequantized scores. Simulated and semi-simulated experiments are conducted showing that their algorithm works against two baselines.

**Requested Changes:**

(1) This is critical for acceptance. Please implement the "fool's" baseline that I described above. I haven't thought about this for nearly as much time as you have, so there are likely to be obvious reasons why the above baseline is clearly bad. If so, I request that either (a) you justify why this baseline should not even be tried or (b) fix it and add its results to the paper. If you have an idea for a simple-but-reasonable baseline that you think makes more sense, please implement that one instead.

(2) Try "expertise-varying" simulations as described in the preceding section. Not required for acceptance, but recommended.

(3) Add a new line to (some of) the error plots that shows the proposed algorithm under the oracle $\lambda$. This would strengthen the conclusions of Figure 4 by further showing that your $\lambda$ selection scheme only pays a small penalty. Not required for acceptance, but recommended.

**Strengths And Weaknesses:**

Strengths
------------
(1) Clear writing

(2) Good motivation

(3) Realistic setting

(4) Experiments are a good start

Weaknesses
----------------
(1) It would be good to have one more stronger baseline. The two you have are essentially "no algorithm at all" and "we set $\lambda$ to infinity even though our $\lambda$-selection procedure wouldn't have chosen it". Although it is not a given that these two would be worse than your proposed algorithm, it is not surprising. What about something extremely simple such as:

If a reviewer gives both paper $A$ and $B$ the same score $r$, but ranks $A$ > $B$, then replace the scores assigned by $A$ and $B$ with $r + c \cdot \verb|np.random.randn()|$ and $r - c \cdot \verb|np.random.randn()|$, respectively. The constant $c$ could be cross-validated in the same you're doing now with $\lambda$. The random number maybe should be the same one and not generated twice. This algorithm will (a) break ties and (b) take into account rankings, (c) is really easy to implement, but it's sloppy and has no theory behind it, so (hopefully) your algorithm will be superior.

(2) The first simulation has papers assigned at random to reviewers. In reality, there is some affinity between papers and reviewers ("expertise") which is taken into account when papers are assigned. This could be expressed as different levels of noise $\sigma$, where a "good" paper-reviewer match is low $\sigma$ and a "bad" paper-reviewer match is a high $\sigma$ and perhaps even a biased score where $y_{rp}$ is drawn from a Gaussian with mean $x^*_p + b$ where $b$ is the bias.



Questions
------------
Figure 2(a): what happens if you extend the x-axis to $\sigma = 2$? Does BRE-adjusted eventually do better than proposed? If so, why?

(2) Section 5.4, "Since the reviewers are all anonymous in this dataset, we generate the reviewer assignments in a random manner." Is this for when you vary the number of papers per reviewer? Please clarify in the text.

---

> ### Author Response · Authors · 2022-07-21
> **Authors' response to Reviewer J9y1 (Part I / III)**
>
> We thank the reviewer for their time and effort in reviewing our paper and providing useful feedback, and the appreciation of our motivation and problem setting. In the following, we use italics when quoting the reviewer’s comment before our response.
>
> **Requested changes**
>
> (1) *“Please implement the ‘fool's’ baseline that I described above.”*
>
> Reference: *“If a reviewer gives both paper A and B the same score r, but ranks A > B, then replace the scores assigned by A and B with r + c*np.random.randn() and r - c*np.random.randn(), respectively. The constant c could be cross-validated in the same you're doing now with \lambda. The random number maybe should be the same one and not generated twice. This algorithm will (a) break ties and (b) take into account rankings, (c) is really easy to implement, but it's sloppy and has no theory behind it, so (hopefully) your algorithm will be superior.”*
>
>
> Thank you for proposing another baseline method. We have implemented this additional baseline and obtained results on both synthetic and ICLR 2017 data. We present our findings in the following three sub-points. We would first kindly remind that the current baseline algorithm, BRE-adjusted-scores (Algorithm 3 in the paper), also satisfies the criterions “a) break ties, (b) take into account rankings”, but we do agree that the ‘fool's’ baseline adds another element of randomness by having a different random number for each reviewer. We thereby refer to this ‘fool's’ baseline as the “randomized baseline”.
>
> 1. **A brief summary of findings**:
> Compared to the current best-performing baseline, BRE-adjusted-scores, the randomized baseline effectively **reduces the number of ties in output scores** by using a different $\xi_r$ for each reviewer r, but it does so without using any additional useful information. The differences in {$\xi_r$}s are entirely due to randomness. As a result, **the randomized baseline does not improve the ranking error compared to BRE-adjusted-scores**. In contrast, our proposed algorithm leverages both rankings and quantized scores, as well as the factor of reviewer-consensus. **Our proposed algorithm remains dominant in the ranking error**, in the multiple experiment settings described below.
>
> 2. **Algorithm description**:
> Note that since Reviewer J9y1 only describes the behavior of the randomized baseline under a simple setting where each reviewer reviews two papers and gives a pairwise comparison, we will first generalize it to our setting, where each reviewer (potentially) reviews >2 papers and gives a partial ranking of reviewed papers. We make sure that under the simple setting, the randomized baseline behaves as described by Reviewer J9y1. The randomized baseline is implemented in a similar way as the BRE-adjusted-scores, but replacing the small constant $\epsilon$ in BRE-adjusted-scores with the value $c * \xi_r$  for each reviewer r. As suggested by Reviewer J9y1, c is the hyperparameter and $\xi_r$ is a reviewer-dependent random number. c is selected using the same QV procedure as \lambda in the proposed algorithm.
>
> 3. **Numerical Results:**
> We present experiment settings and numerical results in this sub-point. Consistent with the paper, we use (normalized) Kendall-tau ranking error (Section 5.1) as the performance measure and compare the methods on both synthetic data (Section 5.3) and ICLR 2017 data (Section 5.4).
> In both datasets, c is selected from a linear grid from [0.025, 0.1], and the random numbers $\xi_r$ are drawn i.i.d from uniform distribution U[0,1], such that the value $c * \xi_i$ remains small and similar to \epsilon=0.05 in BRE-adjusted-scores. For each setting, we run the experiment 10 times with different random seeds.
>
>     1. Synthetic data. We varied the number of papers assigned to each reviewer within {2, 4, 6}. Results are displayed in the table below.  Percentages inside brackets are the percentages of ties in output dequantized scores. In all tables in this response, the column “quantized scores” is omitted because its errors are much larger than the others.
>
>     | Num.papers per reviewer | Proposed algorithm | Randomized baseline | BRE-adjusted-scores |
>     |---|:---:|:---:|:---:|
>     | 2 | $0.042 \\pm 0.003 (1.38\\%)$ | $0.056 \pm 0.003 (9.47\\%)$ | $0.056 \\pm 0.003 (9.53\\%)$  |
>     | 4 | $0.039 \\pm 0.003 (0.52\\%)$ | $0.049 \\pm 0.003 (5.40\\%)$ | $0.049 \\pm 0.003 (5.93\\%)$ |
>     | 6 | $ 0.037 \\pm 0.003 (0.34\\%)$ | $0.044 \\pm 0.003 (3.48\\%)$ | $0.044 \\pm 0.003 (4.38\\%)$ |
>
>     In all three settings, our proposed algorithm outperforms all baselines.
>
> **Please continue to Part II of the response.**

---

### Review · Reviewer_Mh5n · 2022-05-23

**Summary Of Contributions:**

This paper studies the practical problem of generating de-quantized scores in the peer review process, so that area chairs can make more informed decisions. The paper proposes an algorithmic approach to integrate (partial) ranking, provided by each reviewer, into the de-quantized score generation process. The proposed algorithm is a constrained convex optimization problem that can be solved by existing solvers, so the main contribution of this work is the formulation (not an algorithm), which follows three design principals around consistency and consensus. The paper describes its connection to existing models as special cases (e.g., when only scores are available). Experiments are conducted in two synthetic settings: 1) synthetic dataset where data is generated under specific statistical models. 2) "real-world" ICLR2017 dataset. The proposed method can outperform a naive baseline (observed score without any modification, and an adjusted version of an existing method).

**Requested Changes:**

TMLR explicitly asks reviewers to evaluate papers via"Are the claims made in the submission supported by accurate, convincing and clear evidence?". The reviewer feels the paper needs more work to fully meet the criterion:

1. The paper does not really experiment on real-world datasets. The ICLR dataset was artificial in several aspects 1) The scores were used as groundtruth, and another level of quantization was used to generate "observed scores". 2) Reviewer assignments were random. The reviewer understands the difficulty of getting a more ideal dataset, but as a scientific work those difficulties should probably be mitigated nevertheless. Or the authors should elaborate on why and how such artificial settings transfer to the real-world.

2. Besides sythetic dataset, only ICLR2017 dataset is used. There are several datasets in the Kang et al. paper. Please validate the mehods on more datasets or elaborate on why using such datasets is hard. One difficulty the reviewer can see is, only ICLR provides level-10 rating, which allows further artificial quantization and feasible tuning of the hyperparameter (see next point). In summary, only one artificial dataset may not suffice as convincing evidence. The authors are suggested to provide more convincing results.

3. Please compare with more baselines or elaborate on why other baselines are not feasible.

4. One very important hyperparameter in this work is lambda as in Algorithm 1. Since there is no learning in this paper (i.e., existing solver will be applied directly, lambda is pretty much the only knot that controls the solution). This hyperparameter can not be tuned on validation dataset which does not exist, so this paper proposes a QV method, which basically does another level of quantization. There are a couple of concerns that the authors should provide more evidence on why the solution is feasible and practical: a) the other level of quantization clearly requires the original score range not to be small. For example, many conferences only have 5 levels of scores (most scores will be around borderline) and another level  of quantization will likely to lose too much information to make the selection valid. Please elaborate on this point. b). The further quantization is under specific quantization assumptions - please elaborate on why and when the assumptions can transfer to real-world settings.

**Strengths And Weaknesses:**

Stregnths
1. The studied problem has value in the peer review process.
2. The formulation is sound under intuitive principles.
3. The discussion in terms of connection to existing methods and analysis of special cases contribute to the understanding of the framework.

Weakness
1. The major concern is in terms of experiments. TMLR explicitly asks reviewers to evaluate papers via"Are the claims made in the submission supported by accurate, convincing and clear evidence?". The reviewer feels the paper does not fully meet the criterion on several aspects:
a. The paper does not really experiment on real-world datasets.
b. Besides sythetic dataset, only ICLR2017 dataset is used.
c. Only one non-trivial baseline is compared with.
2. One very important hyperparameter in this work is lambda as in Algorithm 1. The selection process of lambda is somehow concerning.

---

> ### Author Response · Authors · 2022-07-21
> **Authors’ response to Reviewer Mh5n (Part I / II)**
>
> We thank the reviewer for their time and effort in reviewing our paper, providing helpful feedback, and their appreciation of our problem formulation and analysis. In the following response, we use italics when quoting the reviewer’s comment.
>
> (1) *“The paper does not really experiment on real-world datasets. The ICLR dataset was artificial in several aspects 1) The scores were used as groundtruth, and another level of quantization was used to generate ‘observed scores’. 2) Reviewer assignments were random. The reviewer understands the difficulty of getting a more ideal dataset, but as a scientific work those difficulties should probably be mitigated nevertheless. Or the authors should elaborate on why and how such artificial settings transfer to the real-world.”*
>
> We understand the concern of the reviewer on the artificial parts of the ICLR 2017 data, **given these concerns, we will change our description of the ICLR 2017 data from “real-world” to “semi-synthetic dataset based on review scores from the ICLR 2017 conference”**.
> We only perform this semi-simulation step to generate rankings and ground truth for dequantized scores, because datasets with both rankings and ratings are not available in raw peer review data.
>
> For the reviewer assignment concern, since the reviewer assignment data is not public, we conducted a new experiment with the following clustered reviewer assignments as a middle ground. We divide the papers in the ICLR 2017 dataset randomly into several clusters, and a reviewer can only review papers in the same cluster. The clustered structure simulates different genres of papers, for example, the Theory of ML, Deep learning, etc… We choose the number of clusters to be 4, and tried two options for cluster sizes: one where all clusters are of approximately the same sizes and another one where cluster sizes are uneven. Within each cluster, we assign papers to reviewers randomly.
> The (normalized) Kendall-tau ranking errors (Section 5.1) are displayed below. The “randomized baseline” method is a new baseline we implemented according to the suggestion of Reviewer J9y1, for details of this new baseline, please see point (1) of our response to Reviewer J9y1. In all the tables below, the column “quantized scores” is omitted because its errors are much larger than the others. Our algorithm remains dominant in the performance.
>
> 1.Similar cluster sizes:
> | Num.papers per reviewer | Proposed algorithm | Randomized baseline | BRE-adjusted-scores |
> |---|:---:|:---:|:---:|
> | 2 | $0.044 \\pm 0.002$ | $0.062 \pm 0.003$ | $0.062 \\pm 0.003$  |
> | 3 | $0.030 \\pm 0.002 $ | $0.042 \\pm 0.002$ | $0.042 \\pm 0.002$ |
> | 6 | $0.010 \\pm 0.001$ | $0.014 \\pm 0.001$ | $0.014 \\pm 0.001$ |
>
> 2.Uneven cluster sizes (in approximate proportion {4:2:1:1}):
> | Num.papers per reviewer | Proposed algorithm | Randomized baseline | BRE-adjusted-scores |
> |---|:---:|:---:|:---:|
> | 2 | $0.043 \\pm 0.002$ | $0.061 \pm 0.002 $ | $0.061 \\pm 0.002 $  |
> | 3 | $0.030 \\pm 0.002 $ | $0.042 \\pm 0.002 $ | $0.041 \\pm 0.002 $ |
> | 6 | $0.010 \\pm 0.002 $ | $0.014 \\pm 0.002 $ | $0.014 \\pm 0.002 $ |
>
>
> (2) *“Besides synthetic dataset, only ICLR 2017 dataset is used. There are several datasets in the Kang et al. paper. Please validate the methods on more datasets or elaborate on why using such datasets is hard. … ”*
>
> There are indeed several other datasets in the cited paper but these are not suitable for our purposes. These datasets are 1. arXiv 2007-2017; 2. NeurIPS 2013–2017; 3. ACL 2017; 4. CoNLL 2016. We elaborate one by one on why these datasets are not suitable.
> 1.  arXiv. There are **no review scores** available for arXiv submissions
> 2.  NeurIPS. Only review texts and the confidence scores for accepted papers are available. Furthermore, **no review scores** are available.
> 3. ACL and 4. CoNLL. The original scores in these two datasets are on a scale of 1 to 5. Please note that the proposed algorithm (including choosing lambda by QV) does run on data with 5 levels of quantization, and the algorithm works well on such data, as supported by our results on ICLR 2017 data, in which the observed scores have only 5 levels. The only caveat is that, if we additionally want to **evaluate its performance** by using the original scores as ground truth as we did on the ICLR 2017 data, we would not be able to do so on the ACL and CoNLL data due to the small number of levels in their original scores.
>
> **Please continue to Part II of the response.**

---

> > ### Author Response · Authors · 2022-07-21
> > **Authors’ response to Reviewer Mh5n (Part II / II)**
> >
> > (3) *“Please compare with more baselines or elaborate on why other baselines are not feasible.”*
> >
> > We implemented another baseline that (a) breaks ties and (b) takes into account rankings, as suggested by Reviewer J9y1. This baseline adds another element of randomness and hence incurs fewer ties in the output than the BRE-adjusted-scores baseline. Despite that, the proposed algorithm outperforms the baselines consistently across different experiment settings. For details, please refer to point (1) in our response to Reviewer J9y1.
> >
> > (4) *“One very important hyperparameter in this work is lambda as in Algorithm 1. …  This hyperparameter can not be tuned on validation dataset which does not exist, so this paper proposes a QV method, which basically does another level of quantization. There are a couple of concerns that the authors should provide more evidence on why the solution is feasible and practical: a) the other level of quantization clearly requires the original score range not to be small. For example, many conferences only have 5 levels of scores (most scores will be around borderline) and another level of quantization will likely to lose too much information to make the selection valid. Please elaborate on this point. b). The further quantization is under specific quantization assumptions - please elaborate on why and when the assumptions can transfer to real-world settings.”*
> >
> > (a) We would like to emphasize that the ICLR 2017 dataset (Section 5.4) has quantized scores in 5 levels (obtained from the 10-level original scores which we treated as ground truth in order to evaluate the algorithms). The experiment results on the ICLR 2017 dataset (Figure 3), show that the proposed algorithm (including choosing lambda with QV) performs well when there are 5 levels of quantized scores. Furthermore, on the ICLR 2017 dataset, Figure 4 (e)-(f) in the paper reveal that the QV process selected a good $\lambda$ that is close to the optimal $\lambda$, the optimal $\lambda$ being the one that achieves the smallest error on the dataset.
> >
> >
> > (b) Using QV to select hyperparameter $\lambda$ (on a further quantized dataset) does not require any additional quantization assumptions beyond what the proposed algorithm requires. Even though our quantization assumption is stated as values in [z-0.5,z+0.5] mapping to integer z (Section 3), all that the proposed algorithm (and therefore QV) requires is that the quantization bins are non-overlapping. This is because the performance measure is the Kendall-tau ranking error (Section 4.2), since our algorithm is arbitrary-scale (Section 4.2). Therefore, the quantization assumption works no matter what level of quantization is used in the data, and hence, the proposed algorithm can directly be applied to both original and further quantized datasets. Please feel free to clarify more if you have additional concerns.
> >
> > Thank you for taking time to read this response.

---

### Review · Reviewer_Bb4m · 2022-08-09

**Summary Of Contributions:**

This paper proposed an algorithm that could help Area Chair (AC) to make better decisions about the acceptance of research papers (referred to as The Algorithm here after). It turns the quantized score and/or partial ordering of papers within each reviewer into a single dequantized score. The resulting score is easier to understand in AC's workflow, since they don't have to consider both a score and partial ordering.

The authors designed the algorithm to respect reviewer consent, be consistent with known partial ordering, and keep the resulting score as close to the original score as possible. Those principles are reflected in the design of the loss function. Finding the best score is a convex optimization problem where a single best solution is guaranteed and computationally tractable.

The authors compared the proposed algorithm with several lines of previous work, notably with BRE and Trunstonian models. Analysis showed that under certain constraints it agrees with the two reference models.

To estimate the hyperparameter lambda without labeled data, the authors proposed a Quantization Validation method, which takes current quantized scores as the truth, and quantizes them more to create a validation set.

The authors used a few synthetic data sets and ICLR 2017 data to evaluate the proposed algorithm with Kendall-tau ranking error. The results showed that the proposed algorithm performs better than the baseline (BRE-adjusted) under various factors (number of reviewers per paper etc). It outperforms baseline on ICLR data too, although by a smaller margin. Further evaluation of Quantization Validation showed that it can find the near optimal hyperparameter.


**Broader Impact Concerns:**

The Reviewer doesn't have any concern about the potential negative impact of the paper. It's limited to the problem of the research community, and the methodology doesn't show any ethical or societal concerns.



**Requested Changes:**

1) Page 4 Paragraph 3: The Reviewer doesn't understand the example "a reviewer giving a strong accept to two papers...". Could you explain more?

2) Page 7 Paragraph 1: The discussion around principle 3 is too brief. Some examples would help the readers.

3) Page 11, Paragraph 2: ".. values of lambda in [Q]uantization validation" should the Q be in lower case?

4) Page 11, Section 5.2, 2: "We make simple adjustments to the baseline.." it's better to briefly summarize the nature of the adjustments to save the readers a trip to the appendix.

5) Page 12, Paragraph 4: Why do we want to look at the L2 error? It's better to give motivation.

6) Page 13 Section 6: It's hard for The Reviewer to understand this sentence: "it should not inadvertently reveal the review information of paper(s) outside their scope".



**Strengths And Weaknesses:**

S1: The paper is very well written, easy to follow and covers the important issues well. The visualizations are carefully plotted to emphasize key information, e.g. the "zoom-in" in Figure 2 (c). The Reviewer really appreciates the efforts.

S2: The paper studied a very practical problem in paper reviews. Examples given are intuitive and convincing. However, The Reviewer hasn't served as an AC yet, so the assessment of the problem is based on intuition.

S3: Good analysis. The paper compared the proposed algorithm with reasonable baselines (BRE/Trunstonian) both in theoretical analysis and experiments. The paper also discussed the convex nature and the complexity of the solution, which helps avoid randomness.
S4: Experiment setup. It's a hard problem to evaluate something without ground truth, but the paper did a reasonably good job to provide useful data.

W1:  The goal is less clearly defined. On one handl, the task lacks true labels, and the authors deliberately said they won't assume the existence of true labels. On the other hand, the goal is defined vaguely as "seamlessly assist the human experts" (Section 3). It's not clear what a good result really means.

For other minor questions, please see Requested Changes.

---

> ### Author Response · Authors · 2022-08-23
> **Authors’ response to Reviewer Bb4m (Part I / II)**
>
> We thank the reviewer for their time in reviewing our paper, and their appreciation of our problem identification and analysis. In the following, we use italics when quoting the reviewer’s comment before stating our response.
>
> **Weakness**
>
> *W1: ”The goal is less clearly defined. On one hand, the task lacks true labels, and the authors deliberately said they won't assume the existence of true labels. On the other hand, the goal is defined vaguely as "seamlessly assist the human experts" (Section 3). It's not clear what a good result really means.”*
>
> Our goal is to incorporate the ranking information into the quantized scores. In other words, we design an algorithm that takes both partial rankings and discrete scores (Section 2) as inputs.  The fact that we do not assume the existence of some true objective quantity scores leads to our focus on **obtaining one output per every reviewer-paper pair**, instead of per every paper (Section 4.1). Finally, the exact form of said output is a **dequantized and continuous score**, so that the Chairs can still perform tasks that they conventionally perform on the original quantized scores (such as sorting) and hence be able to use the same interface (Figure 1) as before, and that is what we meant by "seamlessly assist the human experts" in Section 1 and Section 4.1.
>
> **Requested Changes**
>
> *Page 4 Paragraph 3: The Reviewer doesn't understand the example "a reviewer giving a strong accept to two papers...". Could you explain more?*
>
> In this example, the “strong accept” refers to a high (quantized) score such as 8 out of 10, and a “strong reject” refers to a low score such as 3 out of 10. We meant to say that the model in the cited paper in that paragraph treats the quantized scores as partial rankings, which is modeled as “a list of comparisons between“ elements u,v for all pairs of elements (first paragraph, page 2, Alion 2010). Modeling quantized scores as partial rankings loses important information which is the exact score values themselves. In other words, if there are only 2 papers X, Y and a reviewer gives a score of 8 to paper X and a score of 3 to paper Y, the model will only retain the information that “score to paper X > score to paper Y” and discard the values “8” and “3”.   We have made the same clarification in revised paper.
>
> *Page 7 Paragraph 1: The discussion around principle 3 is too brief. Some examples would help the readers.*
>
>
> We would like to point to the two paragraphs above Principle 3, which contain motivation (the first paragraph) and an example (the second paragraph) for Principle 3. We paste these two paragraphs below for the reviewer’s convenience. Please feel free to indicate any confusion about these paragraphs and we would be happy to make further clarifications.
>
> “The above design principles leave us with many possible choices for $\hat{y}\_{rp}, (r, p) \in A$. For instance, consider a scores-only setting where we observe the quantized score $z\_{r,p}$. For a reviewer-paper pair $(r, p) \in A$, the quantized score $z\_{r,p}$ defines only the range of possible values for the dequantized score $\hat{y}_{r,p}$. Based on only the principles we have established so far, all output values $\hat{y}\_{rp} \in [z\_rp − 0.5, z\_rp + 0.5]$ are equally good. Consequently, we establish another design principle to break ties within this interval.
>
> We use the consensus value among reviewers as the additional signal to break ties for quantized scores. The consensus value for a paper is defined as the average of the scores provided by the reviewers for that paper. Based on only the information from the quantized scores given by other reviewers for that paper, it is natural to envisage that the dequantized score lies closer to the consensus value than away from it. In other words, the consensus signal indicates part of the interval $[z\_{r,p} −0.5, z\_{r,p} + 0.5]$ in which the dequantized score should lie. For example, without any ranking information, if a reviewer r gave a score of 7 and all the other reviewers gave 7 as well for paper p, then the consensus signal from other reviewers offers no additional information within the interval [6.5, 7.5]. However, if the other reviewers all gave quantized scores 4, then their consensus signal indicates that the dequantized score $y\_{r,p}$ should lie within $[4, 7] \\cap [7 − 0.5, 7 + 0.5]$, which is $[6.5, 7]$.”
>
> *Page 11, Paragraph 2: ".. values of lambda in [Q]uantization validation" should the Q be in lower case?*
>
> We thank the reviewer for pointing out the inconsistencies. We have fixed them and are using “Quantization Validation” consistently as the name of the approach in the revised paper.
>
> Please refer to Part II to continue reading this response.

---

> > ### Author Response · Authors · 2022-08-23
> > **Authors’ response to Reviewer Bb4m (Part II / II)**
> >
> > *Page 11, Section 5.2, 2: "We make simple adjustments to the baseline.." it's better to briefly summarize the nature of the adjustments to save the readers a trip to the appendix.*
> >
> > We thank the reviewer for this suggestion. We have added a brief explanation in Section 5.2 in the revised paper. The text we added is “At a high level, in the ICLR 2017 dataset, the partial ranking can be viewed as a total ranking over (tied) groups of papers. Therefore, the adjustment to the baseline (Algorithm 3) is that the output scores are now adjusted from the quantized scores in groups instead of individually.”.
> >
> > *Page 12, Paragraph 4: Why do we want to look at the L2 error? It's better to give motivation.*
> >
> > Our main focus is on the ranking error, and we focus on Kendall-tau as it is perhaps the most widely used notion of the ranking error. As we briefly mention in the paper, our analysis pertaining to the L2 error is auxiliary, and is meant to only provide some additional evidence that although our algorithm is arbitrary-scale, its output behaves reasonably in exact values as well. We choose L2 since it is most commonly used for measuring differences between values.
> >
> > *Page 13 Section 6: It's hard for The Reviewer to understand this sentence: "it should not inadvertently reveal the review information of paper(s) outside their scope".*
> >
> > We thank the reviewer for pointing out that this sentence is not clear. We provide a more detailed explanation below, and we also add this explanation to the discussion section (Section 6) of the revised version.
> >
> > When reviewers provide rankings over their assigned papers, such as in the ICML 2021 conference, the Area Chairs only observe rankings between papers **that they are handling**. For example, an AC might see a partially-obscured ranking (from some reviewer) of this form: A > * > B, where A, B are papers under this AC is handling and * is a paper C that the AC does not handle whose information is therefore hidden from the AC. However, in our algorithm, the dequantized scores of A and B might depend on the score of paper C (and scores given by other reviewers who reviewed paper C, etc.). For example, if A gets a dequantized score of 6 while B gets 3.95, C might have a tie with B with dequantized score of 4.05. Therefore, the dequantized scores may contain more information than the quantized scores plus the partially-obscured rankings revealed to the AC. Although it is unlikely that the AC would be able to directly infer, for example, the identity of reviewers of paper C, it is a potential concern that some information about paper C would be revealed from the dequantized scores given to the AC.
> >
> > Thank you for the time in reading this response.

---

### Review · Reviewer_Dx6m · 2022-08-10

**Summary Of Contributions:**

This paper aims to integrate the ranking information into the scores in the peer review process, with the goal of mitigating the arbitrariness in conference reviews.  They evaluate the proposed approach with synthetic datasets and the ICLR 2017 conferences, and it shows that the proposed approach can largely reduce errors (30%).

**Broader Impact Concerns:**

This algorithm might be related to the fairness of peer review.

**Requested Changes:**

- Adding more experiments with larger datasets.
- Adding experiments on other domains.



**Strengths And Weaknesses:**

### Strengths
- The investigated issue is interesting and important for the peer review process. Ranking information is indeed not well-leveraged. Combining rankings and quantized scores should generate more reliable review results.
- They formulated the problem mathematically and converted it into a convex optimization problem. They also analyzed the special cases.
- They conducted experiments on real-world datasets and demonstrated the effectiveness of the proposed approach. Detailed analyses are given to study the impact of hyperparameter selection.

### Weaknesses
- Nowadays, we usually have thousands of submissions to machine learning related conferences each year. The tested datasets are too small. Also, they only tested the algorithm in the machine learning field. It would be more interesting if experiments on more domains were explored. As the authors mentioned in the broader impact statement, there are many other problems that may affect the quality of the updated scores.  To ensure robustness, more experiments are expected.
- I am also concerned about the fairness of the algorithm. It would be better if there were some analysis on that.

---

> ### Author Response · Authors · 2022-08-23
> **Author’s response to Reviewer Dx6m (Part I / II)**
>
> We thank the reviewer for their time in reviewing our paper, and their appreciation of the problem investigated and the analysis. In the following, we use italics when quoting the reviewer’s comment before stating our response.
>
> **Weakness / Requested changes**
>
> (1) *”The tested datasets are too small.” / “Adding more experiments with larger datasets”*
>
> We considered all datasets in Kang et al., 2018 besides the ICLR 2017 dataset (Kang et al., 2018 is dedicated to presenting such datasets “of scientific peer reviews available for research purposes”) but those are not suitable for our purposes. Below, we clarify one by one why these datasets are not suitable. Please feel free to indicate if the reviewer has any datasets in mind as well.
>
> These datasets are 1. arXiv 2007-2017; 2. NeurIPS 2013–2017; 3. ACL 2017; 4. CoNLL 2016.
> 1.  arXiv. There are **no review scores** available for arXiv submissions
> 2.  NeurIPS. Only review texts and the confidence scores for accepted papers are available. Furthermore, **no review scores** are available.
> 3. ACL and 4. CoNLL. The original scores in these two datasets are on a scale of 1 to 5. Please note that the proposed algorithm (including choosing lambda by QV) does run on data with 5 levels of quantization, and the algorithm works well on such data, as supported by our results on ICLR 2017 data, in which the observed scores have only 5 levels. The only caveat is that, if we additionally want to **evaluate its performance** by using the original scores as ground truth as we did on the ICLR 2017 data, we would not be able to do so on the ACL and CoNLL data due to the small number of levels in their original scores.
>
> Despite the difficulties in collecting real-world peer review datasets, to address the reviewer’s concern about the number of papers in the dataset, we conduct additional experiments with the synthetic data where we increase the number of papers to 2000. The other parameters are set to default (Section 5.3). We present the results below. In all settings, the normalized Kendall-tau ranking errors (Section 5.1) remain consistent and the proposed algorithm outperforms baseline methods. We have revised the paper with these results added to Appendix A.5.
>
> | Total number of papers | Proposed algo | BRE-adjusted-scores | Quantized scores |
> |---|:---:|:---:|---|
> | 200 | $0.0388 \\pm 0.0011$ | $0.0474 \\pm 0.0013 $  | $0.0580 \\pm 0.0011$ |
> | 500 | $0.0384 \\pm 0.0006 $ | $0.0470 \\pm 0.0006$ | $0.0576 \\pm 0.0007$ |
> | 1000 | $0.0385 \\pm 0.0005 $ | $0.0470 \\pm 0.0005 $ | $0.0576 \\pm 0.0005$ |
> | 2000 | $0.0388 \\pm 0.0005 $ | $0.0473 \\ pm 0.0005 $ | $0.0575 \\pm 0.0004 $ |
>
> (2) *”It would be more interesting if experiments on more domains were explored.”/ ”Adding experiments on other domains.”*
>
> We did not fully understand this comment. Could the reviewer please clarify what domain they are thinking of, and let us know if they have any specific relevant datasets they have in mind. We understand that the spirit of the question is to consider other settings for experiments that may be reflective of other domains. To this end, we conducted the following additional experiments with varying parameters, reviewer assignment patterns, etc., that could simulate other domains: (i) experiments with larger numbers of papers (Appendix A.5 of revised paper). (ii) experiments with larger number of papers assigned to each reviewer, where we fix other parameters and increase the number from 2~6 (Section 5.3) to 10. We did the same for the number of reviewers per paper as well. Results for this point are presented in the table below. Our proposed algorithm still outperforms the baselines in these settings. (iii) experiments with varying “reviewers’ expertise” on the synthetic data, details are presented in our response Part II (2) to Reviewer J9y1 and added to Appendix A.6 of the revised paper. (iv) experiments with a clustered reviewer assignment on the ICLR 2017 data, where a reviewer can only review papers from a subset of all paper, details are presented in our response Part I (1) to Reviewer Mh5n and added to Appendix A.7.
>
> | Num.paper per reviewer| Proposed algo | BRE-adjusted-scores | Quantized scores |
> |---|:---:|:---:|---|
> | 10 | $0.0333\\pm 0.0018$ | $0.0375 \\pm 0.0021 $  | $0.0595 \\pm 0.0034$ |
>
> | Num.reviews per paper | Proposed algo | BRE-adjusted-scores | Quantized scores |
> |---|:---:|:---:|---|
> | 10 | $0.0392\\pm 0.0020$ | $0.0482 \\pm 0.0023 $  | $0.0595 \\pm 0.0033$ |
>
> Please continue to Part II of this response.

---

> > ### Author Response · Authors · 2022-08-23
> > **Author’s response to Reviewer Dx6m (Part II / II)**
> >
> > (3) *”​​there are many other problems that may affect the quality of the updated scores. To ensure robustness, more experiments are expected”*
> >
> > As suggested by Reviewer J9y1, we conduct new experiments where the quality of the match between each paper-reviewer pair is varied, by setting different levels of noise in the Thurstone model. In all settings we have tried, the proposed algorithm retains an advantage over baselines. For details, please see our response (2) to Reviewer J9y1. The results are also added to Appendix A.6 in the revised paper.
> >
> > (4) *”It would be better if there were some analysis on that (fairness of the algorithm).”*
> >
> > We outlined and discussed several potential issues in the discussion section (Section 6) as well as in the Broader Impact Statement, including issues such as global accuracy versus accuracy in some subgroups of papers. Connecting possible solutions to these issues with our problem formulation remains a direction for future research.
> >
> > Thank you for taking time to read this repsonse.

---

### Author Response · Authors · 2022-08-23
**List of changes made to the revised PDF manuscript**

We sincerely thank all the reviewers for their effort in reviewing our paper and for providing valuable feedback. We have responded to the reviews separately and revised our paper based on the feedback received from all the reviewers. Below, we summarize the main changes made in the revised version.
1. Added experiments with larger numbers of papers in Appendix A.5. (Reviewer Dx6m)
2. Added experiments with varying reviewers’ expertise, in form of varying variances in the Thurstone model, in Appendix A.6. (Reviewer J9y1)
3. Added a line for the performance of the proposed algorithm with oracle $\lambda$ in Figure 2(b) and Figure 3(a), as additional supporting evidence that the QV process is able to select a good value for the hyperparameter. (Reviewer J9y1)
4. Added experiments with a clustered reviewer assignment, in addition to the completely random assignment on ICLR 2017 data, in Appendix A.7.  (Reviewer Mh5n)
5. Clarified ambiguous parts in the writing, please see individual reviewer comments and responses for the exact locations.
6. Fixed typos and inconsistencies in the writing, please see individual reviewer comments and responses for the exact locations.

---

### Decision · Action_Editors · 2022-09-07

**Recommendation:** Accept as is

**Comment:**

This paper proposes an algorithm that can help area chairs to make better decisions. The algorithm combines a quantized score of the paper with a partial ordering of the papers into a single de-quantized score that can be acted upon. The problem of computing the de-quantized score is formulated as a convex optimization problem. The proposed method is evaluated on synthetic problems and also on ICLR 2017 reviews.

This paper was well received and is also well written. There were two kinds of major comments:
1. Insufficient baselines and experimental evaluations. The authors addressed these comments by running more experiments and including them in the paper.
2. All experiments are conducted only on one real-world dataset of ICLR 2017 reviews. The authors clearly explained that all other paper review datasets do not have the properties that they seek. The datasets either do not have review scores or the granularity of the scores is insufficient to evaluate the proposed method. Given this, my opinion is that the empirical evaluation is sufficient.

The only thing that I ask the authors for is that they include a paragraph that describes the limitations of the other datasets in the paper. I believe that the readers of the paper will find it interesting. Since this is easy to do, simply by copying the paragraph from the rebuttals, this paper is "accepted as is". Congratulations!